# Species-specific sensitivity to TGFβ signaling and changes to the Mmp13 promoter underlie avian jaw development and evolution

**Spenser S Smith, Daniel Chu, Tiange Qu, Jessye A Aggleton, Richard A Schneider***

Department of Orthopaedic Surgery, University of California, San Francisco, San Francisco, United States

**Abstract** Precise developmental control of jaw length is critical for survival, but underlying molecular mechanisms remain poorly understood. The jaw skeleton arises from neural crest mesenchyme (NCM), and we previously demonstrated that these progenitor cells express more bone-resorbing enzymes including *Matrix metalloproteinase 13* (*Mmp13*) when they generate shorter jaws in quail embryos versus longer jaws in duck. Moreover, if we inhibit bone resorption or *Mmp13,* we can increase jaw length. In the current study, we uncover mechanisms establishing species-specific levels of *Mmp13* and bone resorption. Quail show greater activation of and sensitivity to transforming growth factor beta (TGFβ) signaling than duck; where intracellular mediators like SMADs and targets like *Runt-related transcription factor 2* (*Runx2)*, which bind *Mmp13*, become elevated. Inhibiting TGFβ signaling decreases bone resorption, and overexpressing *Mmp13* in NCM shortens the duck lower jaw. To elucidate the basis for this differential regulation, we examine the *Mmp13* promoter. We discover a SMAD-binding element and single nucleotide polymorphisms (SNPs) near a RUNX2-binding element that distinguish quail from duck. Altering the SMAD site and switching the SNPs abolish TGFβ sensitivity in the quail *Mmp13* promoter but make the duck promoter responsive. Thus, differential regulation of TGFβ signaling and *Mmp13* promoter structure underlie avian jaw development and evolution.

**\*For correspondence:**
rich.schneider@ucsf.edu

**Competing interest:** The authors declare that no competing interests exist.

## Editor's evaluation

The manuscript brings new original findings about developmental mechanisms regulating MMP13 activity and associated bone resorption in avian species. These processes lead to the control of jaw size in a species-specific context, therefore, indicating probable evolutionary significance.

## Introduction

Jaws are among the most highly adapted and modified structures of vertebrates, and they facilitate complex behaviors like feeding, respiration, and vocalization. For this reason, precise developmental regulation of jaw length is crucial for survival (*Schneider, 2015*). By comparing jaw development between white Pekin duck and Japanese quail, we have shown in prior work that neural crest mesenchyme (NCM), which is the embryonic progenitor population that gives rise to the jaw skeleton, employs a variety of stage- and species-specific mechanisms to govern jaw length (*Jheon and Schneider, 2009*; *Fish and Schneider, 2014b*; *Schneider, 2018b*; *Schneider, 2018a*). Duck have much longer jaws compared to those of quail, and during the early migration of NCM from the anterior neural tube,

duck embryos allocate more progenitors to the presumptive jaw region (*Fish et al., 2014c*). Once these NCM populations arrive, their growth trajectories further diverge due to autonomous molecular programs for proliferation and differentiation that are tied to intrinsic rates of maturation and species-specific regulation of multiple-signaling pathways (*Eames and Schneider, 2008*; *Merrill et al., 2008*; *Mitgutsch et al., 2011*; *Hall et al., 2014*; *Ealba et al., 2015*).

A key finding from our studies is the identification of a previously unrecognized developmental mechanism governing jaw length, which is NCM-mediated bone resorption. Quail have higher levels of bone-resorbing enzymes than duck during late stages of jaw development (*Ealba et al., 2015*), including tartrate-resistant acid phosphatase (TRAP) and *Matrix metalloproteinase 13 (Mmp13)*, which is a collagenase secreted by osteocytes and other cell types (*Johansson et al., 1997*; *Sasano et al., 2002*; *Behonick et al., 2007*; *Chen et al., 2012a*). In the jaw skeleton, osteocytes are derived exclusively from NCM (*Le Lièvre, 1978*; *Noden, 1978*; *Helms and Schneider, 2003*). Transplanting presumptive cephalic NCM from quail to duck dramatically elevates expression of bone resorption enzymes and generates chimeric "quck" with shorter quail-like jaws, whereas blocking bone resorption using a bisphosphonate or an MMP13 inhibitor significantly lengthens the jaw (*Ealba et al., 2015*). Likewise, knockdown of *Mmp13* alters jaw growth in zebrafish (*Hillegass et al., 2007*) and affects the shape of craniofacial structures during tadpole development (*Pinet et al., 2019*). Human patients with mutations in *Mmp13* also display jaw size defects (*Kennedy et al., 2005*). Such findings reveal that NCM controls bone resorption and that there is a link between bone resorption, *Mmp13* activity, and jaw length. However, what has remained unclear are the molecular mechanisms that lead to differential regulation of *Mmp13* and the species-specific control of bone resorption in relation to jaw length.

To address this question in the current study, we focus on the transforming growth factor beta (TGFβ) signaling pathway, which is known to mediate bone deposition and resorption, as well as *Mmp13* expression (*Stouffer and Owens, 1994*; *Moses and Serra, 1996*; *Viñals and Pouysségur, 2001*; *Kim et al., 2004*; *Selvamurugan et al., 2004a*; *Fang et al., 2012*; *Crane and Cao, 2014*; *Wu et al., 2016*). TGFβ signaling involves ligands interacting at the plasma membrane with a receptor dimer consisting of type I and type II TGFβ receptors. Upon binding of TGFβ ligand, the type II receptor (TGFβR2) transphosphorylates and activates the type I receptor (TGFβR1), initiating an intracellular signaling cascade involving phosphorylation of SMAD2 and SMAD3. These activated SMADs form a complex with SMAD4, allowing translocation into the nucleus and interaction with SMAD-binding elements and DNA-binding proteins to activate or repress transcription of target genes (*Heldin et al., 1997*; *Dennler et al., 1998*; *Derynck et al., 1998*; *Li et al., 1998*; *Massagué and Wotton, 2000*; *Alliston et al., 2001*; *Derynck and Zhang, 2003*). Inhibiting TGFβ receptor kinases suppresses *Mmp13* expression in vivo (*Dunn et al., 2009*), while TGFβR1 activity positively affects MMP13 and the remodeling of bone (*Dole et al., 2017*). Similarly, other target genes such as *Runt-related transcription factor 2* (*Runx2*), which is a major transcription factor expressed by osteoblasts (*Ducy et al., 1997*; *Komori et al., 1997*; *Karsenty et al., 1999*; *Selvamurugan et al., 2004a*; *Derynck et al., 2008*), can be induced or repressed by TGFβ ligands depending on the levels of exposure and the complement of transcriptional co-factors (*Lee et al., 2000*; *Alliston et al., 2001*; *Selvamurugan et al., 2004b*; *Wu et al., 2016*). Overexpressing *Runx2* in NCM can shorten the jaw (*Hall et al., 2014*), whereas patients with *Runx2* haploinsufficiency can develop an enlarged lower jaw (*Gorlin et al., 1990*; *Jaruga et al., 2016*; *Pan et al., 2017*). *Runx2* is also a known regulator of *Mmp13* (*Enomoto et al., 2000*; *Wang et al., 2004*; *Javed et al., 2005*; *Pratap et al., 2005*; *Selvamurugan et al., 2009*; *Komori, 2010*; ; *Chen et al., 2012b*; *Takahashi et al., 2017*).

To identify mechanisms that control the differential regulation of *Mmp13* and potentially link bone resorption and jaw length, we assay for species-specific expression of ligands, receptors, and effectors of the TGFβ pathway in chick, quail, and duck embryos during key stages of jaw development when bone is being deposited and resorbed. We employ these three birds for several reasons. First, for comparative studies, a three-taxon analysis in which two taxa are more closely related than either are to a third taxon is generally accepted as a robust strategy for making the most parsimonious inferences about evolution (*Nelson and Platnick, 1991*; *Mavrodiev et al., 2019*; *Rineau et al., 2020*). As Galliformes, chick and quail are closely related to each other phylogenetically (separated by around 50 million years) and they diverged from a common ancestor with Anseriformes, which include duck, over 100 million years ago (*Pereira and Baker, 2006*; *Hackett et al., 2008*; *Kan et al., 2010*). Second, chick and quail are more similar in terms of their jaw morphology when compared to duck (*Schneider*

*and Helms, 2003*; *Eames and Schneider, 2008*; *Mitgutsch et al., 2011*; *Smith et al., 2015*). Third, chick is a long-established experimental model system with a well-annotated genome (*Stern, 2005*; *Lwigale and Schneider, 2008*; *Sauka-Spengler and Barembaum, 2008*; *Jheon and Schneider, 2009*; *Fish and Schneider, 2014a*; *Abramyan and Richman, 2018*; *Gammill et al., 2019*), which helps in designing experiments and analyzing data for quail and duck (*Ealba and Schneider, 2013*; *Chu et al., 2020*).

We quantify expression of TGFβ pathway members and observe higher levels in the developing jaws of quail versus those of duck and chick. We perform cell and organ culture experiments to test if these species-specific differences are due to intrinsic differences in sensitivity to TGFβ signaling and to test if inhibiting the TGFβ pathway can reduce bone resorption in the developing jaw. We also assess the effects of *Mmp13* overexpression on jaw length in duck. We then search for molecular mechanisms that may underlie species-specific differences in sensitivity to TGFβ signaling by examining the structure and function of the *Mmp13* promoter in chick, quail, and duck. We discover key differences in the structure of the *Mmp13* promoter involving a SMAD-binding element and single nucleotide polymorphisms (SNPs) near a RUNX2 binding element that distinguish quail and chick from duck. To test if such differences affect transcriptional activity, we generate species-specific reporter constructs with or without these binding elements and SNPs. Overall, our results indicate that multiple levels of gene regulation in the TGFβ signaling pathway mediate *Mmp13* expression, bone resorption, and ultimately species-specific variation in jaw length.

## Results

### Bone resorption and MMP13 levels are species-specific and spatially regulated

To identify coincident areas of bone resorption and MMP13 localization, we performed TRAP staining and immunohistochemistry (IHC) for MMP13 on sections of chick, quail, and duck lower jaws at embryonic stage (HH) 40. We performed trichrome staining on adjacent sections to label areas of bone deposition (*Figure 1A–B, F, K, P and U*; *Figure 1—figure supplement 1A,F*). We observe qualitatively higher levels of TRAP staining in chick and quail lower jaws in all bone regions compared to stage-matched duck lower jaws (*Figure 1C–E*, *Figure 1—figure supplement 1B,G*), which is consistent with our published quantifications of TRAP staining in the avian jaw (*Ealba et al., 2015*). In the chick and quail angular bone, MMP13 protein is elevated compared to similar regions in duck and overlaps directly with TRAP staining (*Figure 1F–H and K-M*, *Figure 1—figure supplement 1A-C*). In the chick and quail dentary bone, areas of TRAP staining overlap with domains of MMP13 and are elevated compared to similar regions in duck; however, the duck dentary has elevated levels of TRAP and MMP13 compared to the angular bone (*Figure 1P–R and U–W*, *Figure 1—figure supplement 1F-H*). MMP13 is not detected in cartilage (i.e. Meckel's cartilage) of the lower jaw skeleton at HH40 (*Figure 1—figure supplement 1K,L*). Overall, we find that TRAP staining and MMP13 levels are coincident, generally higher in chick and quail than in duck, and spatially regulated in duck such that TRAP and MMP13 levels are higher in the dentary bone than in the angular bone.

### TGFβ signaling components are present in the lower jaw and upregulated in quail during key stages of bone resorption

To examine whether higher levels of bone resorption and MMP13 protein expression observed in quail versus duck correlate with differential regulation of TGFβ signaling, we performed in situ hybridization, quantitative PCR (qPCR), and RNA sequencing (RNA-seq). We first assayed for the expression of a ligand (i.e. *Tgfβ1*) and a receptor (i.e. *Tgfβr1*) at HH40. In situ hybridization analyses reveal that *Tgfβ1* and *Tgfβr1* are expressed in domains that largely co-localize with areas of osteoid staining for the angular and dentary bones (*Figure 1F–Y*; and *Figure 1—figure supplement 1A-J*).

We then quantified expression of TGFβ ligands, receptors, effectors, and target genes relative to HH31. When examining expression of TGFβ ligands at HH37, we find an increase in *Tgfβ1* and *Tgfβ3* for chick (2.2-fold, p≤0.05; 2.8-fold, p≤0.0006) and quail (twofold, p≤0.05; 4.8-fold, p≤0.0001). In contrast, we observe no change in duck (*Figure 2A–B*). *Tgfβ2* levels do not change in chick or quail but decrease in duck at HH37 (*Figure 2—figure supplement 1A*; 3.4-fold, p≤0.05). For TGFβ receptors *Tgfβr1*, *Tgfβr2*, and *Tgfβr3* mRNA expression does not change over time in any species, whereas

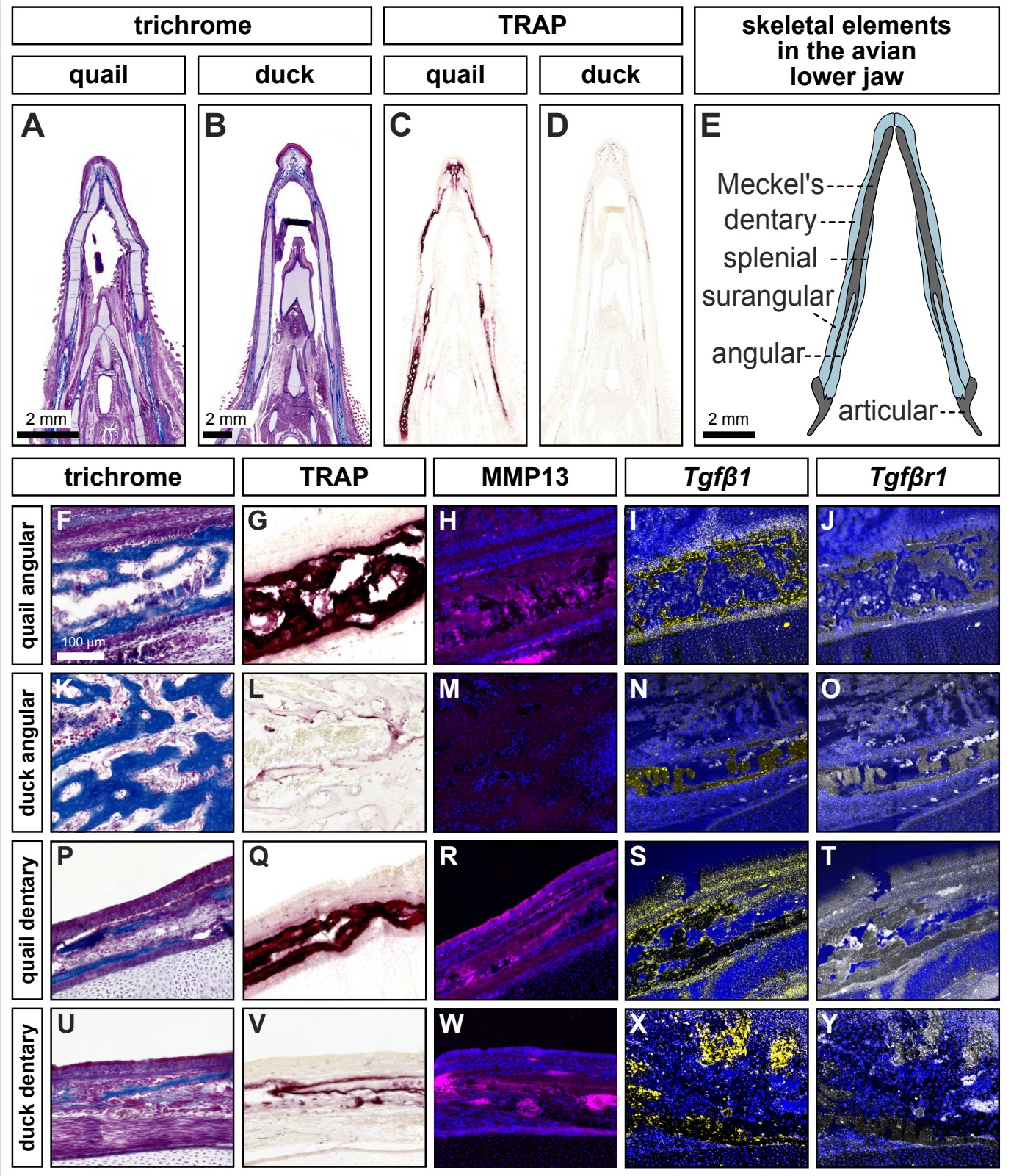

**Figure 1.** Species-specific differences in spatial domains and levels of TRAP, MMP13, *Tgfβ1*, and *Tgfβr1*. Sections of lower jaws stained with trichrome (osteoid matrix of bone is blue) in (**A**) quail (n=4) versus (**B**) duck (n=4) at HH40. (**C**) Adjacent sections stained for TRAP activity (red) reveal more robust bone resorption in quail versus (**D**) duck. (**E**) Schematic of bones within the avian lower jaw. Adjacent sections through the more proximal angular bone stained with (**F**) trichrome, (**G**) TRAP, (**H**) MMP13 antibody (pink) and cell nuclei (blue), (**I**) *Tgfβ1* probe (yellow), and (**J**) *Tgfβr1* probe (white) in

*Figure 1 continued on next page*

*Figure 1 continued*

quail. Angular bone stained with (**K**) trichrome, (**L**) TRAP, (**M**) MMP13 antibody, (**N**) *Tgfβ1* probe, and (**O**) *Tgfβr1* probe reveals substantially less bone resorption in duck. Adjacent sections through the more distal dentary bone stained with (**P**) trichrome, (**Q**) TRAP, (**R**) MMP13 antibody, (**S**) *Tgfβ1* probe, (**T**) and *Tgfβr1* probe in quail. Dentary bone stained with (**U**) trichrome, (**V**) TRAP, (**W**) MMP13 antibody, (**X**) *Tgfβ1* probe, and (**Y**) *Tgfβr1* probe reveals more bone resorption in the dentary versus the angular bone of duck but still less bone resorption overall compared to that observed in quail.

The online version of this article includes the following figure supplement(s) for figure 1:

**Figure supplement 1.** Levels of TRAP activity, MMP13, *Tgfβ1*, and *Tgfβr1* expression in chick.

the non-canonical receptor *Activin a receptor like type 1* (*Acvrl1*) increases at HH37 in quail with no changes in chick or duck (*Figure 2C*, *Figure 2—figure supplement 1B-D*). For downstream effectors, *Smad2* increases at HH37 in chick (2.5-fold, p≤0.0001) and quail (2.8-fold, p≤0.05) but decreases in duck (1.6-fold, p≤0.05), whereas *Smad3* decreases at HH40 in all species (*Figure 2D*, *Figure 2—figure supplement 1E*). We find that at HH37, quail have higher expression of *Tgfβ1* (5.6-fold, p≤0.0004), *Tgfβ3* (3.4-fold, p≤0.009), *Tgfβr1* (6.5-fold, p≤0.001), *Acvrl1* (3.5-fold, p≤0.0001), and *Smad2* (three-fold, p≤0.0001) compared to duck. No differences are found between quail and duck at any stage for *Tgfβ2*, *Tgfβr2*, *Tgfβr3*, and *Smad3*. Expression levels for TGFβ pathway components in chick show similar trends to those observed in quail.

To confirm activation of the TGFβ pathway, we assayed for phosphorylated (p) SMAD3 and observe a fivefold (p≤0.004) higher level in quail versus duck at HH37, indicating that quail have elevated TGFβ signaling at this stage (*Figure 2E*, *Figure 2—figure supplement 2A*, *Figure 2—figure supplement 2—source data 1*). Correspondingly, we find significant upregulation of TGFβ target genes in quail versus duck at HH37 including *Runx2* (4.6-fold, p≤0.0001), *Mmp13* (8.5-fold, p≤0.0001), *Plasminogen activator inhibitor 1* (*Pai1*; 5.3-fold, p≤0.0001), and *Mmp2* (9.3-fold, p≤0.0001; *Figure 2F–G*; supplemental figure S4F-G), as well as MMP13 protein levels (1.6-fold, p≤0.01; *Figure 2H*; *Figure 2—figure supplement 2B*, *Figure 2—figure supplement 2—source data 2*). Gene expression increases between HH34 and HH37 for *Runx2*, *Mmp13*, *Pai1*, and *Mmp2* in quail, which mirrors the increases in TGFβ ligand expression and higher activation of pSMAD3, whereas in duck these genes either decrease or remain flat during the same transition. Chick gene expression follows similar trends to that observed in quail. We also observe an increase in *Mmp9*, which is secreted by osteoclasts (*Reponen et al., 1994*; *Engsig et al., 2000*), in all species at HH37, with higher expression in quail compared to duck (2.3-fold, p≤0.001; *Figure 2—figure supplement 1H*). *Mmp14*, which is mostly secreted by osteocytes (*Wu et al., 2009*; *Qing et al., 2012*; *Dole et al., 2017*), does not change in expression and is similar among species (*Figure 2—figure supplement 3A*). *Cathepsin K* (*Ctsk*) follows a similar trend as other bone resorption markers with increased expression at HH37 in chick (3.2-fold, p≤0.05) and quail (1.5-fold, p≤0.05), but not duck (*Figure 2—figure supplement 3B*). *Sclerostin* (*Sost*) expression increases in chick at HH40 (7.6-fold, p≤0.01) but does not change in quail or duck from HH34 to HH40 (*Figure 2—figure supplement 3C*).

To provide independent confirmation of our qPCR data for ligands and receptors of the TGFβ signaling pathway, we generated a bulk RNA-seq dataset for the lower jaws of chick, quail, and duck at HH37. We calculated read counts for *Tgfβ1*, *Tgfβ2*, *Tgfβ3*, *Tgfβr1*, *Tgfβr2*, and *Tgfβr3*. As with the qPCR data, we find that reads were higher in quail compared to chick and duck for *Tgfβ1* (p≤0.02), *Tgfβr1* (p≤0.003), and *Tgfβr3* (p≤0.003; *Figure 2—figure supplement 4A,D,F*). However, slightly deviating from our qPCR results, *Tgfβ3* expression was higher in chick compared to quail and duck (p≤0.01; *Figure 2—figure supplement 4C*). As with our qPCR data, *Tgfβ2* and *Tgfβr2* reads were not different among any of the species examined (*Figure 2—figure supplement 4B,E*). Read counts for *Mmp13* were also higher in quail compared to duck (*Figure 2—figure supplement 4G*, 10.2-fold, p≤0.001). Taken together, our analyses reveal that certain members and targets of the TGFβ signaling pathway, especially *Tgfβ1*, *Tgfβ3*, *Tgfβr1*, *Smad2*, *Runx2*, *Mmp2*, *Mmp9*, and *Mmp13*, are more highly expressed and show greater activation in chick and quail versus duck during embryonic stages most closely associated with bone resorption in the lower jaw.

## Sensitivity to TGFβ signaling is cell autonomous and species-specific

To test if the greater activation of the TGFβ pathway is related to intrinsic species-specific differences in sensitivity to TGFβ signaling, we performed experiments comparing the response of chick and duck fibroblasts to recombinant (r) TGFβ1 protein. We treated chick and duck cells with rTGFβ1

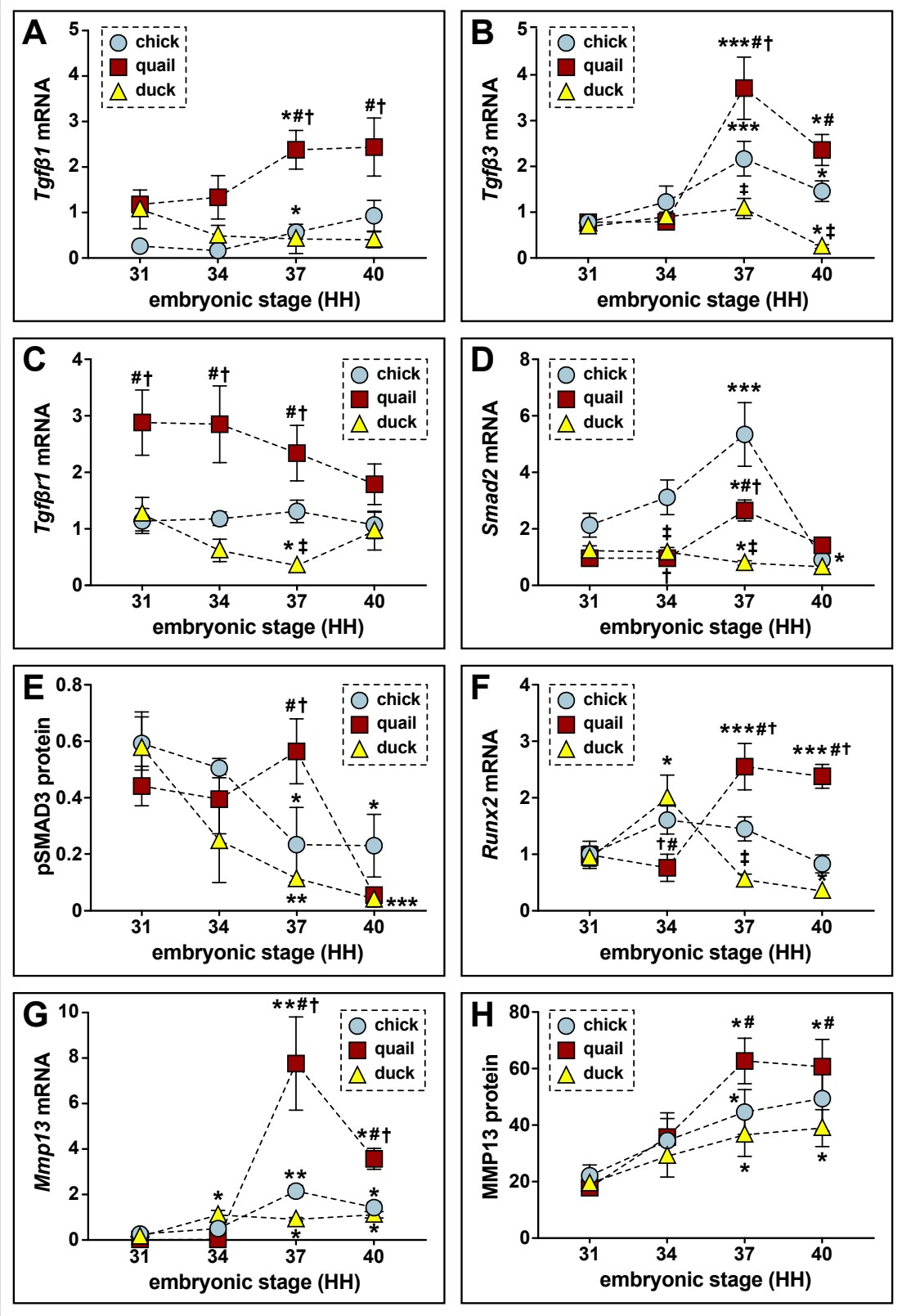

**Figure 2.** Relative mRNA and protein levels of TGFβ pathway members and targets in the developing lower jaws of chick, quail, and duck. (**A**) qPCR analyses show that *Tgfβ1* mRNA increases in chick (blue circles, n=8) and in quail (red squares, n=8) at HH37 and does not change in duck (yellow triangles, n=10) in later developmental stages; *Tgfβ1* is higher in quail than chick or duck at HH37. (**B**) *Tgfβ3* mRNA increases in chick and in quail at HH37, while duck levels decrease by HH40. Chick and quail have higher *Tgfβ3* at HH37 compared to duck. (**C**) *Tgfβr1* mRNA remains constant for chick

*Figure 2 continued on next page*

*Figure 2 continued*

and quail, whereas duck decrease at HH37. Quail maintain higher levels of *Tgfβr1* from HH31 to HH37 relative to chick and duck. (**D**) *Smad2* mRNA increases in chick and in quail at HH37, whereas duck levels decrease. At HH40, chick and quail robustly decrease compared to HH37. (**E**) Western blots show pSMAD3 protein levels are elevated in quail (n=12) at HH37. Duck (n=12) levels decrease over time, while chick (n=9) levels decrease at HH37 and HH40 compared to HH31. Quail have higher levels at HH37 than duck. (**F**) *Runx2* mRNA increases in quail at HH37 and remains elevated at HH40, whereas duck increase at HH34 and then trend downward from HH37 to HH40. Chick *Runx2* does not change. (**G**) *Mmp13* mRNA increases in chick and in quail at HH37, whereas duck increase at HH34 and remains elevated until HH40. (**H**) MMP13 protein increases in chick, in quail, and in duck at HH37 and remains elevated at HH40. p≤0.05 and * denotes significance from HH31 within each group, # denotes significance between quail and duck at same stage, † denotes significance between chick and quail, and ‡ denotes significance between chick and duck.

The online version of this article includes the following source data and figure supplement(s) for figure 2:

**Figure supplement 1.** Relative mRNA levels of TGFβ pathway members and targets in lower jaws of chick, quail, and duck.

**Figure supplement 2.** Representative western blot images.

**Figure supplement 2—source data 1.** Western blot images for phosphorylated SMAD and β-Actin.

**Figure supplement 2—source data 2.** Western blot images for MMP13 and β-Actin.

**Figure supplement 3.** Relative mRNA levels for markers of bone remodeling in the developing lower jaws of chick, quail, and duck.

**Figure supplement 4.** RNA sequencing analysis of TGFβ pathway members in the lower jaws of chick, quail, and duck at HH37.

and assayed for activation of target genes. We observe a threefold (p≤0.001) induction in pSMAD3 protein levels in chick cells treated with rTGFβ1 compared to controls, but no significant response in duck cells (*Figure 3A* , p≤0.07; *Figure 3—figure supplement 1A*, *Figure 3—figure supplement 1—source data 1*). In chick cells, *Runx2* increases threefold at 3 hr (p≤0.02) and 6 hr (p≤0.0005) post-treatment and shows a 7.4-fold induction (p≤0.0001) at 24 hr (*Figure 3B*). However, in duck cells, we find no significant response until 24 hr when we observe a 1.8-fold induction (p≤0.04), which is much less than the induction in chick (p≤0.0001). We observe a similar response in chick cells for *Mmp13* with a 4.2-fold induction (p≤0.0004) at 6 hr and a 10.8-fold induction (p≤0.0001) at 24 hr (*Figure 3C*). In contrast, we only observe a 3.7-fold induction (p≤0.008) of *Mmp13* expression in duck cells at 24 hr. MMP13 protein levels parallel the gene expression response with a 1.6-fold induction (p≤0.001) in chick, but no induction in duck cells (*Figure 3D*, *Figure 3—figure supplement 1B*, *Figure 3—figure supplement 1—source data 2*). We do not observe a similar effect when we examine the response of other targets such as *Pai1* and *Mmp2*. For duck cells treated with rTGFβ1, *Pai1* increases twofold (p≤0.04) at 1 hr and 2.4-fold (p≤0.02) at 24 hr, whereas in chick *Pai1* does not increase until 24 hr with a 3.5-fold induction (p≤0.0001; *Figure 3—figure supplement 2A*). *Mmp2* increases 3.5-fold (p≤0.05) in duck cells at 1 hr and 7.5-fold (p≤0.0001) at 24 hr post-treatment in duck cells, whereas chick have a later and attenuated response with a twofold induction at 6 hr (p≤0.05) and 24 hr (p≤0.05) post-treatment (*Figure 3—figure supplement 2B*). Overall, we observe that chick cells show greater sensitivity to TGFβ signaling than do duck cells, which in turn leads to higher levels of *Runx2* and *Mmp13* expression.

To assess if the observed induction of *Runx2* and *Mmp13* mRNA by rTGFβ1 is mediated via the canonical TGFβ signaling pathway, we utilized small molecule inhibitors of TGFβR1 (i.e. SB431542) and SMAD3 (i.e. SIS3). We treated chick cells with rTGFβ1 and/or SB431542, or rTGFβ1 and/or SIS3 for 24 hr and measured the effects on RUNX2 and MMP13 protein levels. rTGFβ1 treatment induces a twofold increase in RUNX2 (p≤0.0001) and MMP13 (p≤0.0001), whereas inhibitor treatment alone has no effect (*Figure 3E–3F*, *Figure 3—figure supplement 3A-B*, *Figure 3—figure supplement 3—source data 1 and 2*). However, the ability of rTGFβ1 to induce RUNX2 and MMP13 is abolished when cells are treated with rTGFβ1 in combination with either inhibitors of TGFβR1 or SMAD3 (*Figure 3E–F*). We observe no effect on MMP2 protein levels in cells treated with inhibitors of TGFβR1 or SMAD3 (data not shown).

To test if the species-specific differences in sensitivity of chick and duck cells to TGFβ signaling also occur at the organ level and in the context of development, we dissected HH34 lower jaws from chick, quail, and duck (*Figure 4A–C*) and treated them with a series of rTGFβ1 concentrations. We observe no induction of TGFβ pathway members with 5 ng/ml rTGFβ1 treatment in the lower jaws of chick, quail, and duck (data not shown) but observe elevated levels of pSMAD3 in chick (2.3-fold, p≤0.01) and quail (1.8-fold, p≤0.05) versus duck with 25 ng/ml rTGFβ1 treatment for 6 hr (*Figure 4D*, *Figure 4—figure supplement 1A*, *Figure 4—figure supplement 1—source data 1*). For 50 ng/ml, we observe

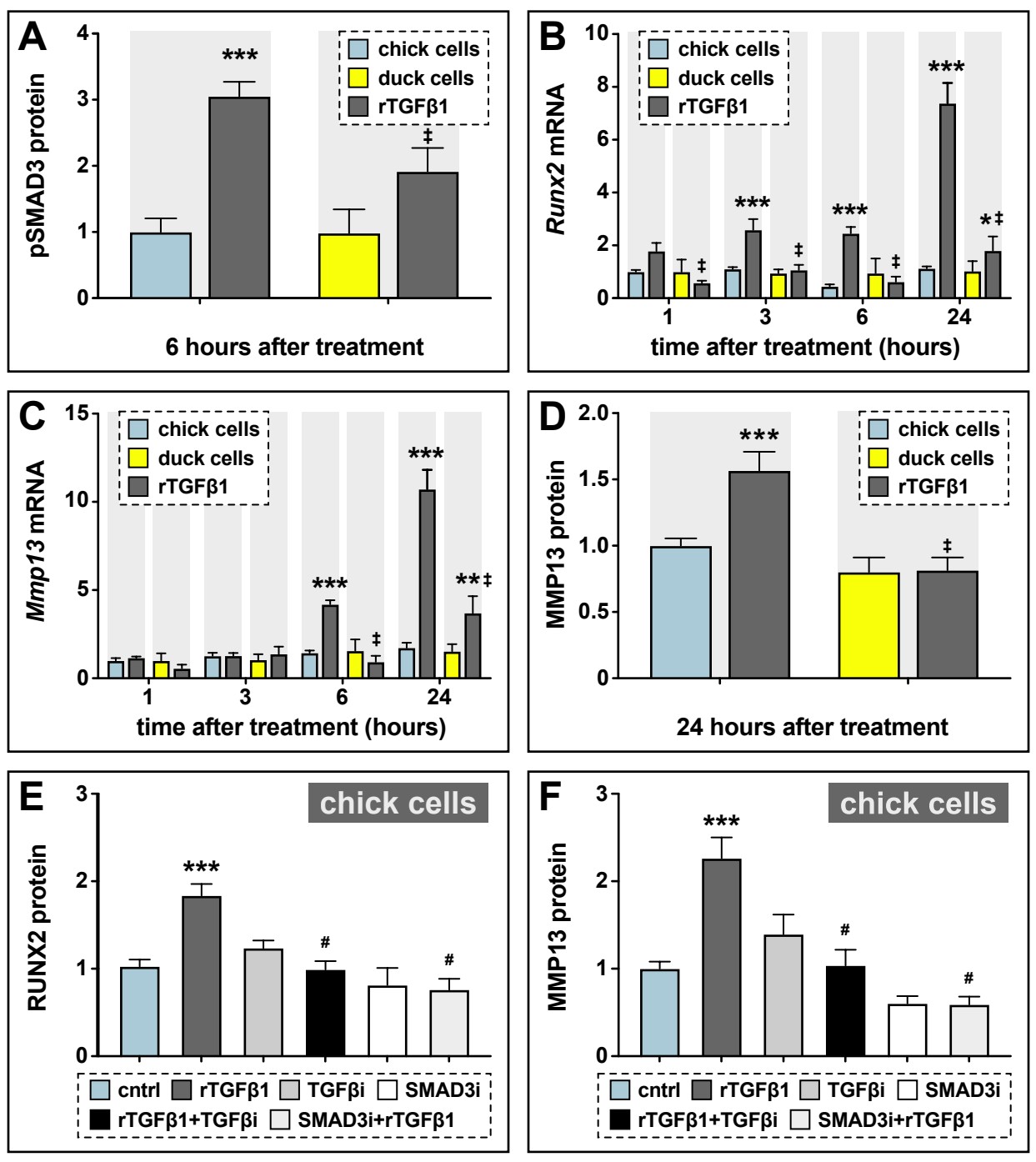

**Figure 3.** Sensitivity of chick and duck cells to rTGFβ1 and effects of TGFβR1 and SMAD3 inhibition on RUNX2 and MMP13. (**A**) pSMAD3 protein levels in cells treated for 2 hr with 5 ng/ml rTGFβ1 (dark gray) in chick (blue) and duck (yellow) cells show a significant induction in chick (n=8). (**B**) Chick and duck cells treated with rTGFβ1 for 1–24 hr. Chick *Runx2* mRNA increases with rTGFβ1 treatment at 3 and 6 hr and at 24 hr, while duck *Runx2* does not increase until 24 hr. In rTGFβ1 treated cells, duck have significantly lower *Runx2* at every time point compared to chick (n=8). (**C**) Chick *Mmp13* mRNA increases with rTGFβ1 treatment at 6 hr and at 24 hr, while duck *Runx2* does not increase until 24 hr. In rTGFβ1 treated cells, duck have lower *Mmp13* at every time point compared to chick (n=8). (**D**) MMP13 protein in cells treated for 24 hr with rTGFβ1 shows an induction in chick but no response in duck (n=8). (**E**) RUNX2 protein levels in chick cells treated for 24 hr with rTGFβ1 (dark gray), TGFβR1 inhibitor (medium gray), a combination of both rTGFβ1 and TGFβR1 inhibitor (black), SMAD3 inhibitor (white), and a combination of both rTGFβ1 and SMAD3 inhibitor (light gray). RUNX2 protein increases with rTGFβ1, but when rTGFβ1 is combined with either a TGFβr1 or a SMAD3 inhibitor, there is a significant decrease compared to rTGFβ1 alone (n=12). (**F**) MMP13 protein increases with rTGFβ1 treatment, but when rTGFβ1 is combined with either a TGFβR1 or a SMAD3 inhibitor, there is a significant decrease compared to rTGFβ1 alone (n=12). * denotes significance from control p≤0.05, ** denotes significance from control p≤0.01, *** denotes

*Figure 3 continued on next page*

*Figure 3 continued*

significance from control p≤0.001, # denotes significance from rTGFβ1, † denotes significance between chick and quail, and ‡ denotes significance between chick and duck.

The online version of this article includes the following source data and figure supplement(s) for figure 3:

**Figure supplement 1.** Representative western blot images.

**Figure supplement 1—source data 1.** Western blot images for pSMAD and β-Actin.

**Figure supplement 1—source data 2.** Western blot images for MMP13 and β-Actin.

**Figure supplement 2.** Sensitivity of chick and duck cells to rTGFβ1 and effects on TGFβ target genes.

**Figure supplement 3.** Representative western blot images.

**Figure supplement 3—source data 1.** Western blot images for RUNX2 and β-Actin.

**Figure supplement 3—source data 2.** Western blot images for MMP13 and β-Actin.

an induction in quail (3.4-fold, p≤0.002), but no response in duck lower jaws. Quail pSMAD3 levels are higher (2.3-fold, p≤0.02) compared to duck for 50 ng/ml. Following treatments with 25 ng/ml rTGFβ1 for 24 hr within species we observe an increase in *Runx2* mRNA expression in chick (2.4-fold, p≤0.03) and quail (3.2-fold, p≤0.0001), as well as in *Mmp13* expression in chick (2.5-fold, p≤0.0001) and quail (3.2-fold, p≤0.0001) but not in duck (*Figure 4E–F*). Relative to duck, treatment with rTGFβ1 increases *Runx2* in chick (3.2-fold, p≤0.0001) and quail (4.3-fold, p≤0.0001) and increases *Mmp13* in chick (five-fold, p≤0.0001) and quail (6.4-fold, p≤0.0001). MMP13 protein levels only show increases in quail at 25 ng/ml, but not in chick or duck (*Figure 4G*, *Figure 4—figure supplement 1B*, *Figure 4—figure supplement 1—source data 2*).

To test if rTGFβ1 treatment affects TGFβ receptor expression, we assayed for *Tgfβr1*, *Tgfβr2*, and *Tgfβr3*, and find no change in expression for any species (*Figure 4—figure supplement 2A-B*, and data not shown). To test if rTGFβ1 treatment stimulates other known TGFβ target genes, we examined *Pai1* and *Mmp2*. *Pai1* mRNA expression in chick and duck does not change with rTGFβ1 treatment, whereas we observe a 1.8-fold induction (p≤0.003) in quail (*Figure 4—figure supplement 2C*). *Mmp2* is induced in chick (1.5-fold, p≤0.02) and quail (1.5-fold, p≤0.04) but not in duck lower jaws (*Figure 4—figure supplement 2D*). To examine if treatment with rTGFβ1 induces markers for bone formation in lower jaws within 24 hr, we assayed for changes in *Collagen type 1 α 1* (*Col1a1*) and *Osteocalcin* (*Ocn*). We do not observe changes in *Col1a1* levels with rTGFβ1 treatment in any species; however, we do observe a 1.8-fold decrease (p≤0.05) in *Ocn* levels in duck (*Figure 4—figure supplement 2E-F*). Overall, we observe greater activation of the TGFβ pathway in the lower jaws of chick and quail versus duck, which may reflect intrinsic species-specific differences in sensitivity to TGFβ signaling.

## Bone resorption requires TGFβ signaling in the jaw skeleton

To determine the extent to which members and targets of the TGFβ pathway regulate bone resorption, we dissected HH35 lower jaws from quail; treated them on the right side with beads soaked in inhibitors of TGFβR1 (*Figure 5A*), SMAD3 (*Figure 5B*), or MMP13 (*Figure 5C*); cultured them in osteogenic media for 5 days, and then performed TRAP staining. Vehicle control beads were implanted on the left side of each sample. Our results show that the area of TRAP staining around the bead is decreased in quail lower jaws by 28% following TGFβR1 inhibition (p≤0.0004), 33% following SMAD3 inhibition (p≤0.009), and 36% following MMP13 inhibition (p≤0.02) when compared to the area of TRAP staining around the control bead on the contralateral side (*Figure 5D–F*; *Figure 5—source data 1*). Thus, bone resorption in the lower jaw skeleton requires TGFβ signaling.

## MMP13 overexpression decreases lower jaw length in duck

In previous work, we demonstrated that treating quail embryos with an MMP13 inhibitor can lead to an increase in lower jaw length (*Ealba et al., 2015*). To test if overexpressing *Mmp13* can also alter jaw length, especially in duck embryos, which show the lowest endogenous levels of *Mmp13* (*Figure 1M*; *Figure 2G and H*), we employed our stably integrating and doxycycline (dox)-inducible overexpression construct (*Chu et al., 2020*). To validate the enzymatic activity of overexpressed MMP13 protein, we used a fluorometric method for measuring gelatinase and collagenase activity in cell lysates. Fluorescence signal remained quenched in empty vector controls, whereas we observe highly fluorescent

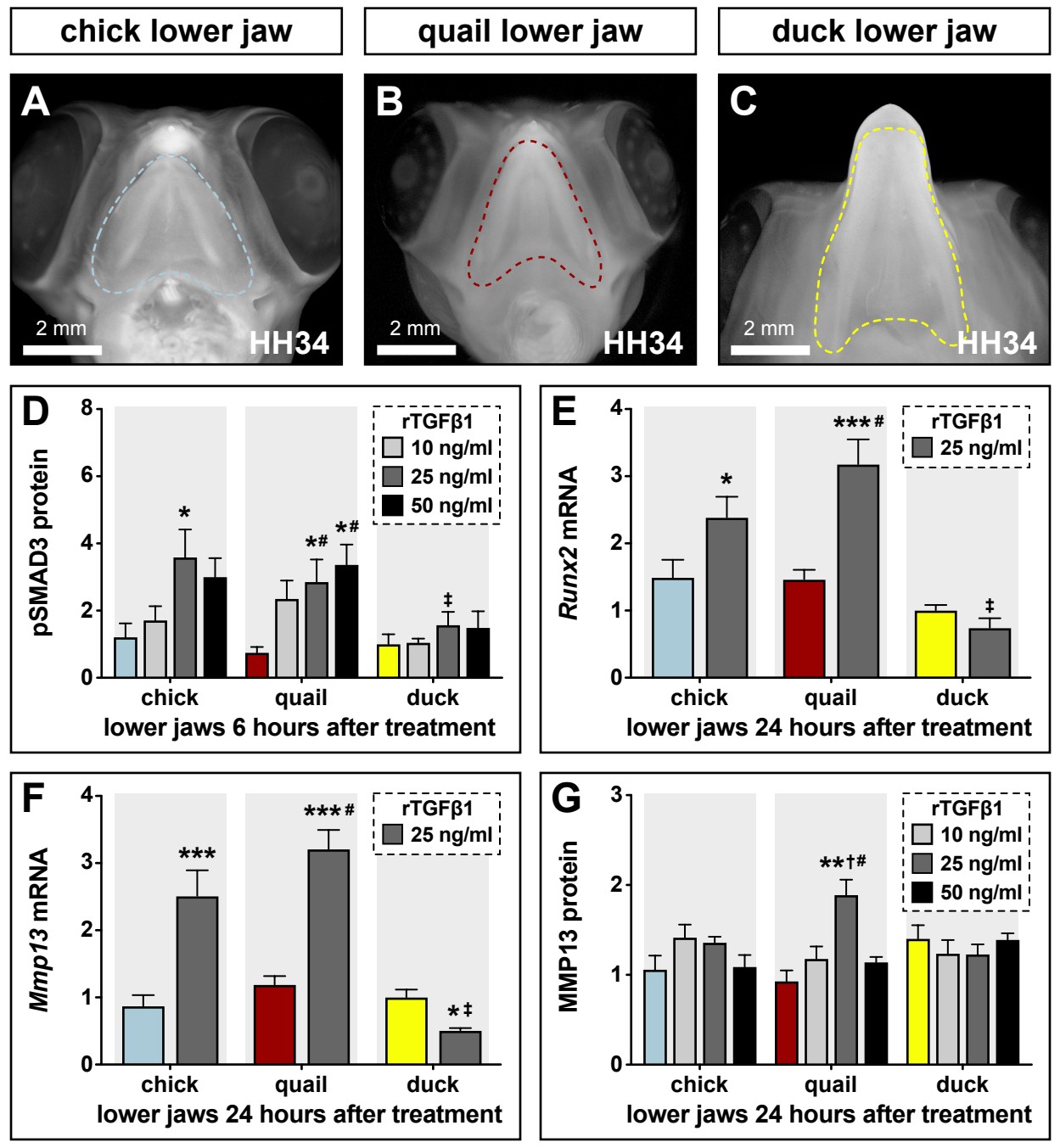

**Figure 4.** Sensitivity of chick, quail, and duck lower jaws to rTGFβ1 and the effects on RUNX2 and MMP13. (**A**) HH34 chick, (**B**) quail, and (**C**) duck heads in ventral view showing the boundaries (dashed line) of the lower jaws dissected for analyses and culture experiments. (**D**) Chick (blue), quail (red), and duck (yellow) lower jaws treated in culture with 10 (medium gray), 25 (dark gray), and 50 (black) ng/ml of rTGFβ1 for 6 hr. pSMAD3 levels increase in chick and in quail at 25 ng/ml, while duck shows no induction. Quail pSMAD3 levels are significantly induced with 50 ng/ml treatment (n=12). (**E**) Chick, quail, and duck lower jaws treated with 25 ng/ml rTGFβ1 for 24 hr. *Runx2* mRNA increases in chick and quail but decreases in duck (n=10). (**F**) Chick, quail, and duck lower jaws treated with 25 ng/ml rTGFβ1 for 24 hr. *Mmp13* increases in chick and quail but decreases in duck (n=10). (**G**) Chick, quail, and duck lower jaws treated in culture with 10, 25, and 50 ng/ml of rTGFβ1 for 24 hr. MMP13 protein levels show an induction in quail with 25 ng/ml rTGFβ1 but no response in chick or duck. * denotes significance from HH31 within each group p≤0.05, ** denotes significance from HH31 within each group p≤0.01, # denotes significance between quail and duck at same stage, † denotes significance between chick and quail, and ‡ denotes significance between chick and duck.

The online version of this article includes the following source data and figure supplement(s) for figure 4:

*Figure 4 continued on next page*

*Figure 4 continued*

**Figure supplement 1.** Representative western blot images.

**Figure supplement 1—source data 1.** Western blot images for pSMAD and β-Actin.

**Figure supplement 1—source data 2.** Western blot images for MMP13 and β-Actin.

**Figure supplement 2.** Sensitivity of chick, quail, and duck lower jaws to rTGFβ1 and effects on TGFβ target genes and *Tgfβr* expression.

fragments indicative of enzymatic digestion following *Mmp13* overexpression (*Figure 5—figure supplement 1A*).

Duck embryos were electroporated bilaterally with either pPIDNB-*Mmp13* or empty vector at HH8.5, treated with dox at HH35, and collected at HH40 for subsequent microcomputed tomography (µCT) analyses. To confirm overexpression, we screened embryos for red fluorescence within the lower jaw as an indication of the extent of dox induction in NCM-derived tissues (*Figure 5G*). µCT was used to generate 3D models of the lower jaw (*Figure 5H1*; *Figure 5—source data 2*), landmarks were annotated on the surface of each 3D model (*Figure 5—figure supplement 1B-C*), and the total distance was measured and compared between control and treated lower jaws. In duck embryos overexpressing *Mmp13,* we observe an approximately 2 mm decrease in lower jaw length (p≤0.004), which represents about 10% of the total length (*Figure 5J*). Thus, the regulation and expression of *Mmp13* affect the growth and length of the lower jaw.

## Species-specific response to TGFβ signaling is mediated by the Mmp13 promoter

To determine the extent to which the *Mmp13* promoter itself may regulate the differential and species-specific response of *Mmp13* to TGFβ signaling, we sequenced a 2 kb fragment immediately upstream from the transcriptional start site of the chick, quail, and duck *Mmp13* promoters (*Figure 6A*). Chick cells were transfected with either the duck or quail *Mmp13* promoter attached to luciferase. To assess the effects of TGFβ signaling on the regulation of the *Mmp13* promoter, cells were treated with rTGFβ1, a TGFβR1 inhibitor, or a combination of both. For the duck *Mmp13* promoter, we find that treatment with rTGFβ1 has no effect on promoter activity, inhibiting TGFβR1 decreases promoter activity (2.1-fold, p≤0.0001), and a combination of rTGFβ1 and TGFβR1 inhibition decreases promoter activity (*Figure 6B*; 2.4-fold, p≤0.0001). In contrast, for the quail *Mmp13* promoter, we find a twofold activation (p≤0.0001) following treatment with rTGFβ1 but no change in promoter activity with TGFβR1 inhibition. No change in promoter activity is observed with a combination of rTGFβ1 and TGFβR1 inhibition compared to untreated cells, but activity is lower compared to rTGFβ1 treated cells (2.2-fold, p≤0.0001). Additionally, to assess the effects of *Runx2* on the regulation of the *Mmp13* promoter, we co-transfected cells with –2 kb of the duck or quail *Mmp13* promoters attached to luciferase, as well as a species-specific *Runx2* overexpression construct (*Chu et al., 2020*). We find that overexpressing duck *Runx2* has no effect on the duck *Mmp13* promoter; however, overexpressing quail *Runx2* induces the activity of the quail *Mmp13* promoter by 2.8-fold (*Figure 6C*; p≤0.0001). Thus, TGFβ signaling and *Runx2* appear to regulate the *Mmp13* promoter in a species-specific manner.

## Differential regulation of Mmp13 is due to species-specific promoter elements

To identify potential regulatory mechanisms underlying these species-specific differences in sensitivity to TGFβ signaling and in levels of *Mmp13* expression, we compared the structure and function of the *Mmp13* promoter in chick, quail, and duck. We mapped potential transcription factor-binding elements and uncovered several species-specific differences, but we focused primarily on binding elements for SMADs and RUNX2 (*Figure 6A*; *Figure 6—figure supplement 1*). Across the –2 kb promoter fragment, we find five SMAD-binding elements in chick, four in quail, and three in duck; and three RUNX2-binding elements in chick, quail, and duck. These binding elements are distributed across the *Mmp13* promoter at locations that are distinct to each species (*Figure 6A*, *Supplementary file 4*). Additionally, one of the SMAD-binding elements that we find present in the chick and quail proximal promoter but absent in duck (*Figure 6D*) is not annotated in the JASPAR 2020 database (i.e. 5′-GGC(CG/GC)–3′) and has previously been shown to be regulated by SMAD3 (*Martin-Malpartida et al., 2017*). We also identified another possible SMAD-binding motif nested entirely

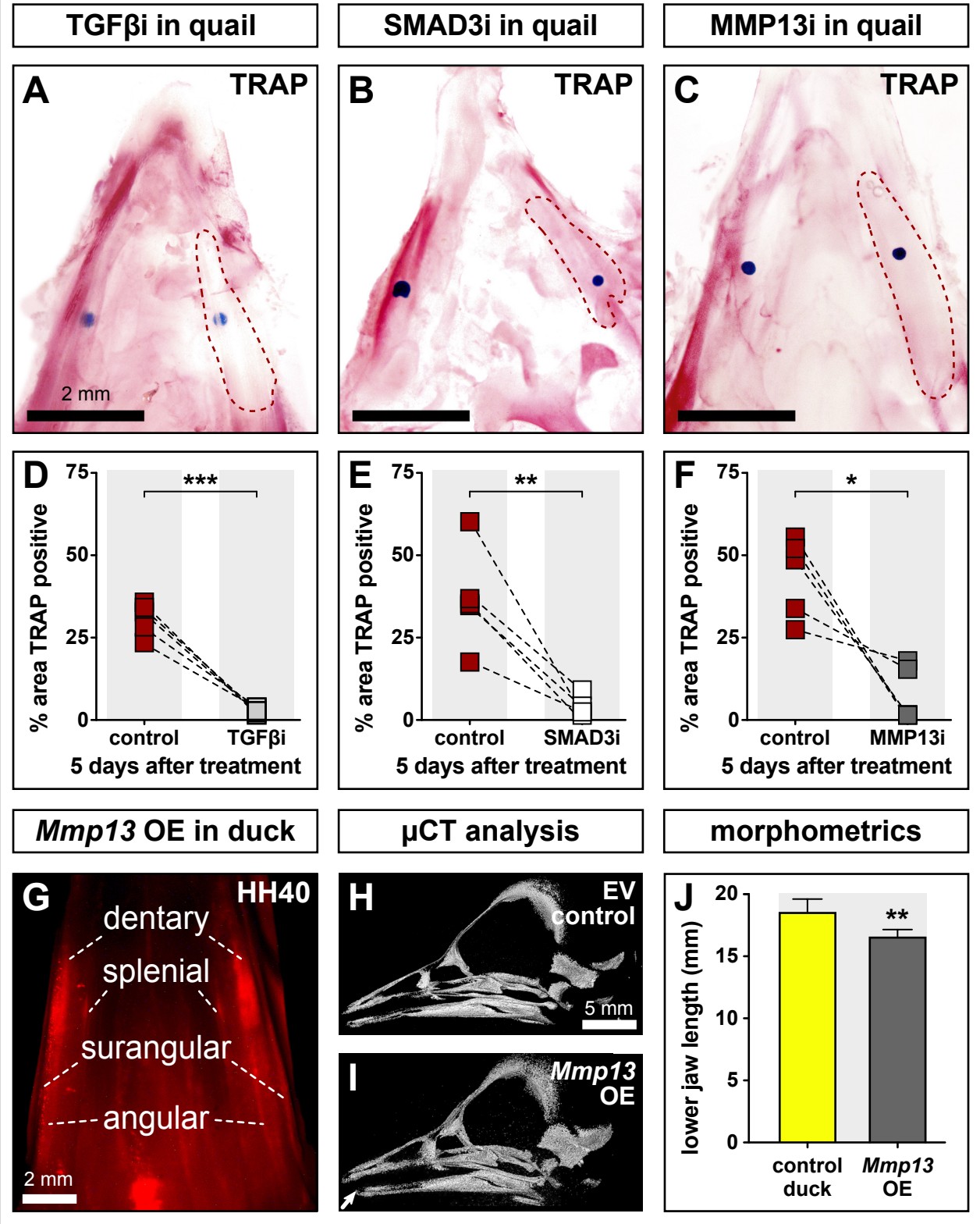

**Figure 5.** Effects of TGFβR1, SMAD3, and MMP13 inhibition on TRAP staining and effects of MMP13 overexpression (OE) on jaw length. Quail lower jaws harvested at HH35 and placed in culture with control beads (left side) and treatment beads (right side) soaked in (**A**) TGFβR1 inhibitor (TGFβi, n=5), (**B**) SMAD3 inhibitor (SMAD3i, n=5), and (**C**) MMP13 inhibitor (MMP13i, n=5). The effects of inhibitor treatments can be seen on TRAP staining (red) after 5 days of culture (red dashed lines). Quantification of the % area of TRAP-positive staining for (**D**) TGFβR1 inhibitor (medium gray), (**E**) SMAD3 inhibitor (white), and (**F**) MMP13 inhibitor (dark gray) in comparison to the contralateral control side. (**G**) In ovo electroporations of the pPIDNB-*Mmp13* OE

*Figure 5 continued on next page*

Figure 5 continued

construct were performed at HH8.5 and embryos were allowed to develop until HH35 when they were treated with a single dose of dox. Electroporation efficiency and extent of OE were evaluated during embryo collection at HH40 by detecting RFP (red). Microcomputed tomography (µCT) analysis of duck specimens electroporated with (H) empty vector or (I) *Mmp13* OE construct. A 3D model was annotated with landmarks from which (J) lower jaw distances were calculated. Embryos overexpressing *Mmp13* showed significantly shorter jaws compared to empty vector controls. * denotes significance from control within each group p≤0.05, ** denotes significance from control within each group p≤0.01, and *** denotes significance from control within each group p≤0.001.

The online version of this article includes the following source data and figure supplement(s) for figure 5:

Source data 1. Effects of TGFβR1, SMAD3, and MMP13 inhibition on TRAP staining in quail embryos.

Source data 2. Effects of MMP13 overexpression on jaw length in duck embryos as visualized by µCT.

Figure supplement 1. Enzymatic activity and morphometrics following *Mmp13* overexpression (OE).

within an activator protein-1 (AP1) binding element in the proximal region of the *Mmp13* promoter. We excluded this potential SMAD site from our experimental design and analysis given the functional importance of AP1 binding for the regulation of *Mmp* expression (*Benbow and Brinckerhoff, 1997*; *Pendás et al., 1997*; *Chakraborti et al., 2003*; *Selvamurugan et al., 2004b*; *Samuel et al., 2007*; *Singh et al., 2010*; *Hashimoto et al., 2013*) and the fact that we could not mutate the shared SMAD motif without simultaneously modifying the AP1-binding element.

To determine the extent to which these SMAD- and RUNX2-binding elements account for the differential and species-specific response of *Mmp13* to TGFβ signaling, we generated four sets of differently sized fragments of the *Mmp13* promoter each driving luciferase expression (*Figure 6A and D*): (1) a –2 kb fragment contained all of the SMAD- and RUNX2-binding elements (as described above); (2) a –184 bp fragment contained one RUNX2-binding element and one SMAD-binding element in chick and quail, but a –181 bp fragment contained only a RUNX2-binding element in duck; (3) a –160 bp fragment had no RUNX2-binding elements but contained a SMAD-binding element in chick and quail, whereas a –157 bp fragment contained no RUNX2- or SMAD-binding elements in duck; and (4) a –1815 bp fragment for chick and quail and a –1818 bp fragment for duck contained all the binding elements upstream of the first −184/181 bp (i.e. ≤−185/182 bp) but without the minimal promoter. We transfected each promoter fragment into chick and duck cells, treated the cells with rTGFβ1 for 24 hr, and measured luciferase activity.

We observe the highest endogenous activity and induction by rTGFβ1 with the –2 kb promoters of chick (1.6-fold, ±p≤0.0001) and quail (1.7-fold, p≤0.0004), whereas the duck –2 kb promoter shows low activity with no response to rTGFβ1 in chick cells (*Figure 6E*). In duck cells (*Figure 6F*), we observe a similar response for the –2 kb promoter of chick (1.4-fold, p≤0.0001) and quail (1.7-fold, p≤0.0004). In chick cells, rTGFβ1 induces the −184 bp chick (1.4-fold, p≤0.05) and quail (1.4-fold, p≤0.02) promoter and produces a similar response in duck cells. In contrast, there is no response to rTGFβ1 observed for the duck −181 bp promoter in either chick or duck cells. For the −160/157 bp promoters, which remove all RUNX2-binding elements from all species, we observe reduced activity for quail and chick, and they do not respond to TGFβ1 treatment. When we examined the −1815/1818 bp fragments, which lie upstream of the first −184/181 bp (i.e. ≤−185/182 bp), we observe reduced or no activity across all species and no response to rTGFβ1. Overall, we find the –2 kb and –184 bp promoter fragments from chick and quail have the highest activity and are strongly induced by rTGFβ1, whereas the duck promoter shows lower activity and is not responsive to treatments with rTGFβ1.

To confirm that the SMAD-binding element present in the proximal *Mmp13* promoter of chick and quail but not duck (*Figure 7A*) differentially binds SMAD protein, we performed an electrophoretic mobility shift assay (EMSA; *Hellman and Fried, 2007*) using recombinant SMAD4 and biotinylated oligos containing the SMAD-binding element sequence of chick/quail or duck. When we raise the concentration of SMAD4, we observe a dose-dependent shift and increase in the binding of chick and quail *Mmp13* promoter oligos (*Figure 7B*). In contrast, despite increasing concentrations of SMAD4, we observe no shift or binding interactions with the duck *Mmp13* promoter oligos. We confirmed the specificity of the interaction between SMAD4 and the binding element present in the chick and quail *Mmp13* promoter by using an identical competitive *Mmp13* oligo that was not biotinylated. We observe a substantial reduction in bound oligo with the addition of competitor oligos (*Figure 7— figure supplement 1A*). Thus, SMAD4 directly interacts with proximal portion of the *Mmp13* promoter in chick and quail but not in duck.

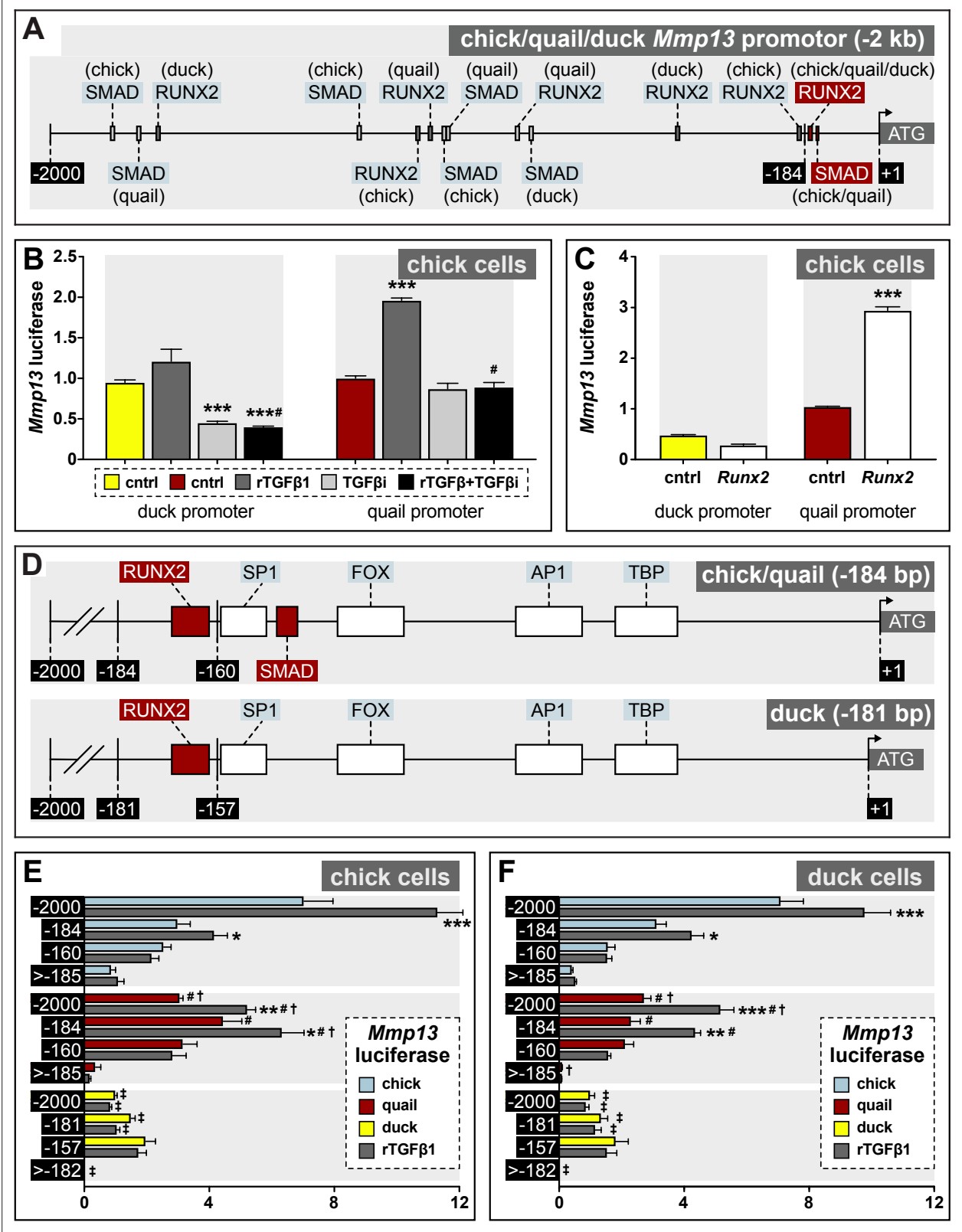

**Figure 6.** Regulation of *Mmp13* via species-specific promoter elements. (**A**) Schematic of a –2 kb region of the chick, quail, and duck *Mmp13* promoter upstream from the transcriptional start site (ATG). There are four SMAD-binding elements in chick, three in quail, and one in duck; and three RUNX2-binding elements in chick, quail, and duck. (**B**) Chick cells transfected with the –2 kb fragment of the duck (yellow) or quail (red) *Mmp13* promoters attached to a luciferase reporter and treated for 24 hr with rTGFβ1 (dark gray), TGFβR1 inhibitor (medium gray), or a combination of both rTGFβ1 and

*Figure 6 continued on next page*

*Figure 6 continued*

TGFβR1 inhibitor (black). rTGFβ1 treatment has little effect on the duck *Mmp13* promoter but induces activity in the quail promoter. TGFβR1 inhibitor as well as a combination of the TGFβR1 inhibitor and rTGFβ1 decreases activity of the duck *Mmp13* promoter. The inductive effects of rTGFβ1 on the *Mmp13* promoter are abolished in quail when rTGFβ1 is combined with TGFβR1 inhibitor (n=18). (**C**) Chick cells transfected with duck (yellow) or quail (red) *Mmp13* promoters plus either an empty vector (cntrl) or with a *Runx2*-overexpression plasmid. *Runx2* overexpression has little effect on the duck *Mmp13* promoter but induces quail *Mmp13* promoter activity (n=18). (**D**) Schematic of the –184/181 bp most proximal region of the chick, quail, and duck promoters. Chick, quail, and duck have similar binding elements for tata-binding protein (TBP), activator protein-1 (AP1), forkhead box (FOX), specificity protein-1 (SP1), and RUNX2. However, chick and quail contain a SMAD-binding element, whereas duck do not. (**E**) Chick and (**F**) duck cells transfected with either a –2 kb fragment; a –184 bp fragment including one RUNX2 and one SMAD-binding element in chick (blue) and quail; a –181 bp fragment including only a RUNX2-binding element in duck; a –160 bp fragment with no RUNX2-binding elements but including a SMAD-binding element in chick and quail; a 157 bp fragment without RUNX2 or SMAD-binding elements in duck; a –1815 bp fragment for chick and quail; or a –1818 bp fragment for duck including all binding elements upstream of the first –184/181 bp (i.e. ≤ –185/182). rTGFβ1 treatment (dark gray) induces activity of the –2 kb and –184 bp fragment for chick and quail; however, no induction is observed in the duck promotor. In the –160 bp fragment, rTGFβ1 induction is abolished in chick and quail. In the 2000–185 bp fragment, very little activity is observed (n=24). * denotes significance from untreated control within each group p≤0.05, ** denotes significance from empty vector control within each group p≤0.01, *** denotes significance from empty vector control within each group p≤0.001, # denotes significance between quail and duck, † denotes significance between chick and quail, and ‡ denotes significance between chick and duck.

The online version of this article includes the following figure supplement(s) for figure 6:

**Figure supplement 1.** Sequence logos for RUNX2 and SMADs.

We also performed an EMSA to confirm that the RUNX2-binding element present in the proximal *Mmp13* promoter of chick, quail, and duck (**Figure 7A**) binds RUNX2 protein. We transfected chick cells with quail or duck pPIDNB-*Runx2*, extracted the nuclear portion, and assayed for binding of biotinylated oligos containing the *Runx2* consensus binding motif of chick/quail or duck. We confirmed the specificity of the interaction by using an identical *Runx2* consensus binding motif competitor oligo that was not biotinylated. For both quail and duck, we observe a substantial reduction in bound *Mmp13* oligo with the addition of *Mmp13* competitor oligos (**Figure 7—figure supplement 1B**). Thus, RUNX2 specifically interacts with binding elements within the proximal *Mmp13* promoter of chick, quail, and duck.

## SNPs by a RUNX2-binding element affect the species-specific activity of Mmp13

Our comparative analyses of the *Mmp13* promoter identified two SNPs that distinguish chick and quail (i.e. adenine and guanine ('AG')) from duck (i.e. cytosine and adenine ('CA')) directly adjacent to a RUNX2-binding element within the –184/181 fragments (**Figure 7A**). To test if these SNPs affect the differential and species-specific response of the *Mmp13* promoter, we put the chick/quail 'AG' SNPs into the duck promoter and the duck 'CA' SNPs into the chick/quail promoter. In a second set of constructs, we added the SMAD-binding element to the duck *Mmp13* promoter and replaced the SMAD-binding element in the chick/quail promoter with duck sequence. We also made a third set of constructs that added both the chick/quail RUNX2 SNPs and SMAD-binding element to the duck *Mmp13* promoter and vice versa. We cloned each of these –184/181 fragments into a luciferase-expression vector and transfected them into chick and duck cells, which were then treated with rTGFβ1. We find that putting the duck 'CA' SNPs into the chick and quail *Mmp13* promoters lowers activity and abolishes induction by rTGFβ1 (**Figure 7C and D**). Similarly, removing the SMAD-binding element reduces chick and quail promoter activity and diminishes the response to rTGFβ1. Moreover, chick and quail promoters containing both the duck 'CA' SNPs and SMAD-binding element deletion show an equivalent diminished response to rTGFβ1 compared to the –184 fragment treated with rTGFβ1. In contrast, when we put the chick/quail 'AG' SNPs into the duck promoter, we observe an increase in endogenous activity, although the SNP switch alone is not sufficient to recover induction by rTGFβ1. We observe the same effect with increased endogenous activity when we put the SMAD-binding element into the duck promoter but no induction with rTGFβ1. However, when we put both the chick/quail 'AG' SNPs and the SMAD-binding element into the duck promoter, we increase endogenous activity and trigger sensitivity to rTGFβ1, with a twofold induction in chick cells (p≤0.002) and 2.4-fold induction (p≤0.05) in duck cells compared to the untreated promoter.

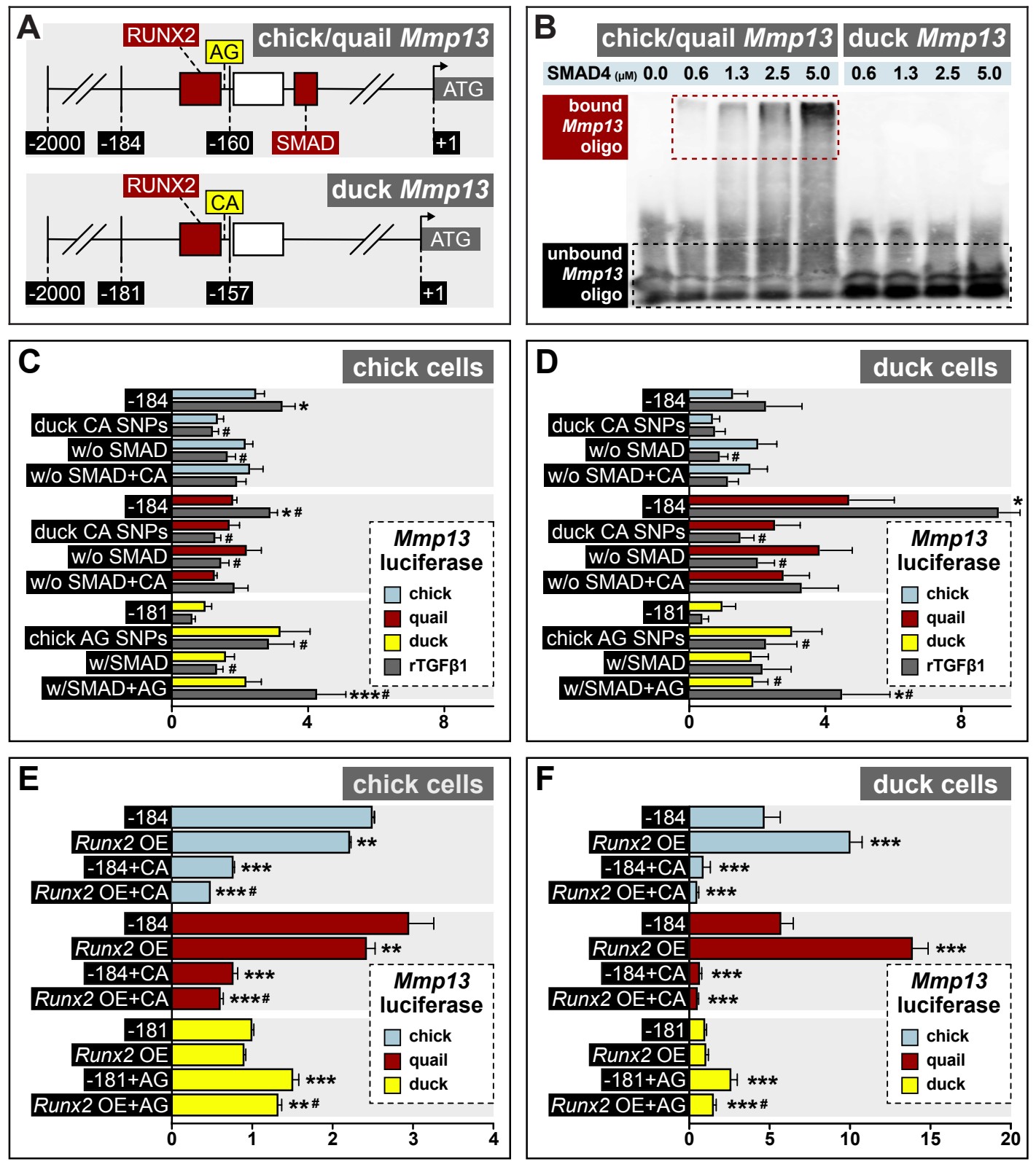

**Figure 7.** Response of the *Mmp13* promoter to rTGFβ1 in relation to species-specific single nucleotide polymorphisms (SNPs) and RUNX2 and SMAD-binding elements. (**A**) Schematic of the proximal region of the chick/quail (–184 bp) and duck (–181 bp) promoters. Chick/quail contain two SNPs (i.e. adenine and guanine ('AG')) adjacent to a RUNX2-binding element that differ from duck (i.e. cytosine and adenine ('CA')). (**B**) Electrophoretic mobility shift assay analysis of the chick, quail, and duck *Mmp13* biotinylated promoter (i.e. –184 or –181) in the presence of increasing concentrations of SMAD4

*Figure 7 continued on next page*

*Figure 7 continued*

recombinant protein. A dose-dependent effect is observed with a shift in bound oligo with increasing concentrations of SMAD4 protein in the chick and quail promoter. However, no shift or interaction is observed with increasing concentrations of SMAD4 protein in the duck promoter. (**C**) Chick and (**D**) duck cells transfected with the chick (blue) or quail (red) *Mmp13* promoter containing either a 184 bp control fragment including the chick/quail 'AG' SNPs plus a RUNX2 and SMAD-binding element; a fragment with the duck 'CA' SNPs plus a RUNX2 and SMAD-binding element; a fragment with the chick/quail 'AG' SNPs plus RUNX2 but no SMAD-binding element; or a fragment including the duck 'CA' SNPs plus RUNX2 but no SMAD-binding element. Chick and duck cells were also transfected with the duck (yellow) *Mmp13* promoter containing either a control fragment including the 'CA' SNPs plus a RUNX2 but no SMAD-binding element; a fragment including the chick/quail 'AG' SNPs plus one RUNX2 but no SMAD-binding element; a fragment including the duck 'CA' SNPs plus one RUNX2 and one SMAD-binding element; or a fragment including the chick/quail 'AG' SNPs plus one RUNX2 and one SMAD-binding element. Cells were treated with rTGFβ1 (dark gray) and assayed for luciferase activity. Switching the chick/quail SNPs to 'CA' SNPs abolishes rTGFβ1 induction. Switching the duck SNPs to 'AG' increases basal activity but does not add a rTGFβ1 response. Removing the chick/quail SMAD-binding element abolishes rTGFβ1 induction. Switching both the RUNX2 SNPs and SMAD-binding element in chick/quail with the duck sequence abolishes rTGFβ1 induction. Switching both the AG SNPs and the SMAD-binding element into the duck promoter adds a response to rTGFβ1 (n=24). * denotes significance from untreated control within each group p≤0.05, ** denotes significance from empty vector control within each group p≤0.01, *** denotes significance from empty vector control within each group p≤0.001, # denotes significance between quail and duck, † denotes significance between chick and quail, and ‡ denotes significance between chick and duck. (**E**) Chick and (**F**) duck cells transfected with the chick, quail, or duck *Mmp13* promoter fragments described in (**B**) and (**C**), in combination with a *Runx2* overexpression (OE) construct. In chick cells, *Runx2* OE decreased *Mmp13* promoter activity for both chick and quail promoters, while no change was observed in the duck promoter. Putting the duck SNPs in chick/quail leads to a decrease in *Mmp13* promoter activity even with *Runx2* OE. Putting the chick/quail SNPs in duck leads to a decrease in promoter activity with *Runx2* OE. In duck cells, *Runx2* OE induced *Mmp13* promoter activity in chick and quail, while no change was observed in the duck promoter. Switching the SNPs in chick/quail to duck abolished the *Runx2* induction. However, switching the SNPs in duck to chick/quail repressed promoter activity with *Runx2* OE (n=24). * denotes significance from the endogenous promoter within each species p≤0.05, ** denotes significance from the endogenous promoter within each species p≤0.01, *** denotes significance from the endogenous promoter within each species p≤0.001, and # denotes significance between control SNP switch and *Runx2* OE with the SNP switch within each species.

The online version of this article includes the following figure supplement(s) for figure 7:

**Figure supplement 1.** SMAD4 and RUNX2 interactions with the chick, quail, and duck *Mmp13* promoter.

To test if the SNPs adjacent to the RUNX2-binding element play a role in the species-specific regulation of the *Mmp13* promoter by RUNX2, we transfected chick and duck cells with the −184 or −181 bp *Mmp13* promoter constructs and with constructs in which we switched the chick/quail 'AG' and duck 'CA' SNPs. These same chick and duck cells were also co-transfected with either an empty vector or a dox-inducible *Runx2* overexpression construct (***Chu et al., 2020***). Our results show that in chick cells, *Runx2* overexpression significantly decreases the activity of the chick and quail *Mmp13* promoter, whereas *Runx2* overexpression has little effect on the duck promoter (***Figure 7E***). In chick cells, replacing the chick/quail SNPs with the duck SNPs decreases activity of the chick (3.2-fold, p≤0.0001) and quail (3.8-fold, p≤0.0001) promoters, and *Runx2* overexpression reduces promoter activity in quail (1.2-fold, p≤0.001) but not chick. Replacing the duck SNPs with chick/quail SNPs induces activity of the promoter by 1.5-fold (p≤0.0001), and *Runx2* overexpression does not change *Mmp13* promoter activity. Similar results are observed in duck cells (***Figure 7F***). Thus, we find that SNPs near a RUNX2-binding element affect the species-specific activity of *Mmp13*.

## Discussion
### MMP13 is co-expressed with TRAP and shows species-specific regulation

In previously published work, we demonstrated that *Mmp13* and bone resorption are regulated by NCM and are important for controlling the species-specific length of the jaw skeleton (***Ealba et al., 2015***). We observed that quail have higher levels of bone resorption markers than do duck during jaw development, including *Mmp13* and TRAP. Transplanting presumptive NCM from quail to duck dramatically elevates expression of *Mmp13* and TRAP and generates "quck" chimeras with shorter quail-like jaws. We also showed that blocking resorption using a bisphosphonate or an MMP13 inhibitor significantly lengthens the jaw, whereas activating bone resorption can shorten the jaw (***Ealba et al., 2015***). In the present study, we identify regulatory mechanisms that lead to the significantly higher levels of *Mmp13* and bone resorption observed in quail relative to duck during development of the lower jaw.

MMP13 is a type 1 collagenase involved in bone resorption and can be produced by osteoblasts, osteocytes, as well as hypertrophic chondrocytes (*Johansson et al., 1997*; *Sasano et al., 2002*; *Hatori et al., 2004*; *Stickens et al., 2004*; *Holmbeck et al., 2005*; *Behonick et al., 2007*; *Tang et al., 2012*; *Zhang et al., 2012*; *Yamamoto et al., 2016*). In the avian jaw skeleton, all these *Mmp13*-expressing cell types are derived entirely from NCM (*Le Lièvre, 1978*; *Noden, 1978*; *Helms and Schneider, 2003*). Although *Mmp13* is expressed by hypertrophic chondrocytes when cartilage is replaced by bone during endochondral ossification (*Colnot and Helms, 2001*), throughout development of the avian lower jaw, Meckel's cartilage persists (i.e. does not undergo hypertrophy), and there is no endochondral ossification except for that limited entirely to the most proximal region within the articular cartilage beginning after HH39 (*Starck, 1989*; *Eames et al., 2004*; *Mitgutsch et al., 2011*; *Svandova et al., 2020*). Therefore, as we have shown previously, we do not detect *Mmp13* mRNA in cartilage of the lower jaw skeleton (*Ealba et al., 2015*) nor do we detect MMP13 protein in the current study. Instead, we observe MMP13 within the bones of the lower jaw, which all form through intramembranous ossification, including the angular and the dentary as they undergo remodeling.

When we examine the quail angular bone, we find elevated MMP13 protein levels coincident with greater amounts of TRAP staining, whereas in the angular bone of duck, we observe very low levels of TRAP and MMP13 despite substantial amounts of osteoid. Within the dentary bone of both quail and duck, however, MMP13 and TRAP are co-expressed, although levels for quail appear elevated compared to duck. This finding indicates there are regulatory mechanisms that can be deployed spatially and control both the species-specific levels and bone-specific domains of MMP13 and TRAP expression, which ultimately may create zones of remodeling that regulate the size and shape of the jaw skeleton. Such a result is consistent with other work proposing that differential fields of resorption underlie changes in the size and shape of the developing human jaw skeleton (*Enlow et al., 1975*; *Moore, 1981*; *Radlanski and Klarkowski, 2001*; *Radlanski et al., 2004*). Given that quail presumptive NCM, when transplanted into duck embryos, autonomously executes molecular and cellular programs not only for resorption, but also for the induction, differentiation, and mineralization of bone during intramembranous ossification (*Merrill et al., 2008*; *Hall et al., 2014*; *Ealba et al., 2015*), our results suggest that NCM more broadly controls pattern in the jaw skeleton through species-specific regulation of TGFβ signaling and especially via differential expression of *Mmp13* and *Runx2* as agents of resorption and deposition that modulate jaw morphology.

## TGFβ signaling mediates species-specific bone resorption in the jaw skeleton

TGFβ signaling is known to play a crucial role in the local osteocyte-mediated remodeling of bone (*Dole et al., 2017*). Thus, we tested if species-specific levels of bone resorption and *Mmp13* expression observed in quail versus duck correlate with differential regulation of the TGFβ pathway. Our in situ hybridization analyses show expression of a TGFβ ligand and receptor in the right place and at the right time to be playing a role, and our parallel strategies for quantifying levels of TGFβ pathway members and targets reveal that the pathway is elevated in the developing jaw primordia of chick and quail relative to duck. For instance, we find higher levels of *Tgfβ1* and *Tgfβr1* in quail and chick versus duck at HH37, which is the stage when bone resorption can first be detected via TRAP staining in the lower jaw (*Ealba et al., 2015*). We also observe greater activation of the TGFβ pathway in quail and chick versus duck based on elevated pSMAD2 and pSMAD3. Additionally, quail show higher expression of target genes including *Runx2*, *Pai1*, *Mmp13*, and *Mmp2*. At a slightly later stage of quail development (i.e. HH40) when bone mineralization and resorption are elevated (*Hall et al., 2014*; *Ealba et al., 2015*), we observe that *Tgfβ3*, *Smad2*, and *Mmp13*, as well as pSMAD3 decrease in expression. This implies that once the TGFβ pathway becomes activated and initiates bone resorption, there is negative feedback to ensure dampened signaling at later stages. Our observation is supported by multiple studies finding both dose and time to be crucial parameters in mediating and determining the effects of TGFβ (*Alliston et al., 2001*; *Takeuchi et al., 2010*; *Chen et al., 2012a*; *Abou-Ezzi et al., 2019*). While TGFβ signaling has been shown to be expressed broadly in craniofacial tissues during avian development (*Yamagishi et al., 1999*; *Cooley et al., 2014*; *Woronowicz et al., 2018*), notably, we do not observe differences in the expression of all TGFβ pathway members among chick, quail, and duck, such as with *Tgfβ2*, *Tgfβr2*, *Tgfβr3*, and *Smad3*. This suggests that the species-specific

activity of the pathway is facilitated by the differential regulation of certain members and targets but not others.

To test for a mechanistic connection between TGFβ signaling and bone resorption, we inhibited the pathway at the level of a receptor (i.e. TGFβR1), intracellular mediator (i.e. SMAD3), and a target gene (i.e. MMP13). We assayed for changes in bone resorption in quail lower jaws via TRAP staining. We find that inhibition at each level of the TGFβ pathway leads to a decrease in bone resorption. This supports our published work showing a direct link between the species-specific amount of bone resorption and jaw length (*Ealba et al., 2015*), and reveals that TGFβ signaling plays an essential role in this process. In other words, the elevated levels of bone resorption observed in the jaw skeleton of quail relative to duck depend upon TGFβ signaling. Taken together, our comparative analysis of the TGFβ pathway in chick, quail, and duck demonstrates that certain ligands, receptors, intracellular mediators, and downstream effectors are upregulated and activated in quail and chick lower jaws when bone resorption is initiated, and this likely serves as a mechanism through which quail and chick achieve higher levels of bone resorption and generate a relatively shorter jaw length than do duck.

## Sensitivity to TGFβ and differential activation of target genes are species-specific

Our study reveals that the developing lower jaw of duck is significantly less sensitive to TGFβ signaling than developing lower jaws of chick and quail, which respond with greater SMAD3 phosphorylation and induction of *Runx2* and *Mmp13* when treated with rTGFβ1. We observe equivalent differences in the species-specific response of duck fibroblasts when compared to chick fibroblasts, indicating that sensitivity to TGFβ signaling is cell autonomous and suggesting that phosphorylation of SMAD3 and the induction of *Runx2* and *Mmp13* in quail versus duck do not depend upon the context of development. This does not mean that duck cells are unable to respond to rTGFβ1, since we observe faster and greater induction of other TGFβ targets including *Pai1* and *Mmp2* when compared to chick cells (although we do not observe such a response in duck lower jaws). We also find that rTGFβ1 has no measurable effect on the expression of bone differentiation markers including *Col1a1* in any species or on *Ocn* in chick and quail. Interestingly, treatment with rTGFβ1 appears to repress *Ocn* in duck, which may reflect the species-specific regulation of osteogenesis. Such results are consistent with previous work demonstrating that the induction or repression of these genes depends upon the stage of differentiation as well as other factors (*García-Trevijano et al., 1999*; *Palcy et al., 2000*; *Gurlek and Kumar, 2001*; *Subramaniam et al., 2001*; *Iwata et al., 2010*; *Chen et al., 2012b*; *Pan et al., 2013*).

Nonetheless, the finding that some pathway targets are responsive to rTGFβ1, and others are not, provides a valuable internal control and indicates that additional regulatory mechanisms are at work that underlie the differential sensitivity to TGFβ signaling and enable the species-specific activation of *Runx2* and *Mmp13* in quail versus duck. Potential mechanisms could involve feedback upon the pathway itself and/or genetic/epigenetic changes to the regulatory landscape especially at the level of target genes. As a proof-of-concept, we explored such possibilities by examining the effects of treatments on TGFβ receptor expression and by focusing on species-specific variation in SMAD- and RUNX2-binding elements in the *Mmp13* promoter. In terms of the effects of TGFβ signaling on receptor expression, we observe no changes in *Tgfβr1*, *Tgfβr2*, or *Tgfβr3* expression in lower jaws treated with rTGFβ1. This suggests that any positive or negative feedback on pathway activation or on levels of receptor expression may occur independent of ligand availability, which differs from what has been observed in some other systems (*Duan and Derynck, 2019*). That being said, our inhibitor experiments further demonstrate that the ability of rTGFβ1 to induce *Runx2* and *Mmp13* occurs directly via activation of TGFβR1 and phosphorylation of SMAD3, which is a finding that has been reported previously (*Selvamurugan et al., 2004a*; *Chen et al., 2012b*; *Chen et al., 2020*). Inhibition of TGFβR1 can also lead to upregulation of *Runx2* and *Mmp13* via a non-canonical p38 MAPK pathway (*Chen et al., 2012a*), which is something we did not analyze in the current study.

Accordingly, one scenario to account for differences in sensitivity to rTGFβ1 between species could be the differential regulation of *Tgfβr1*, since we observe significantly higher levels of expression in quail during three of the four developmental stages analyzed. TGFβR1 is a transmembrane serine/threonine kinase that heterodimerizes with TGFβR2 and is a critical component of TGFβ signal transduction (*Cheifetz et al., 1990*; *Laiho et al., 1990*; *Franzén et al., 1993*; *Tomoda et al., 1994*; *Feng*

*and Derynck, 1997*; *Vellucci and Reiss, 1997*; *Chen et al., 2006*; *Derynck et al., 2008*). TGFβR1 is activated by transphosphorylation once TGFβ ligand binds to TGFβR2, which in turn activates target genes (*Wrana et al., 1994*; *Massagué and Wotton, 2000*). Since we observe constant and comparable levels of *TGFβR2* expression in chick, quail, and duck, and TGFβR1 must heterodimerize with TGFβR2 for signal transduction to occur, then a rate determining step that could serve as a mechanism for modulating pathway activation could be the species-specific regulation of *Tgfβr1*. Determining how *Tgfβr1* is differentially regulated between quail and duck remains a subject of interest for future research. SNPs in *Tgfβr1* and in its promoter are linked to decreased *Tgfβr1* expression as well as craniofacial, skeletal, and other disorders including jaw length defects (*Wu et al., 2002*; *Chen et al., 2004*; *Loeys et al., 2005*; *Pasche et al., 2005*; *Zhao et al., 2008*; *Pasche et al., 2010*; *Sun et al., 2011*; *Knobloch et al., 2019*). Receptor localization may also play a vital role since previous studies have shown that cells can modulate the translocation of TGFβ receptors from the cytosol to their integration at the plasma membrane and allow ligands to bind and initiate signal transduction (*Zhang et al., 1999*; *Rys et al., 2015*; *Duan and Derynck, 2019*). TGFβ receptors can become internalized through clathrin-dependent endocytosis or lipid rafts facilitated by caveolin-1 (*Anders et al., 1998*; *Di Guglielmo et al., 2003*; *Luga et al., 2009*). Once internalized, they can further induce signal transduction, be recycled, or be targeted for degradation (*Mitchell et al., 2004*; *Penheiter et al., 2010*). Additional work could involve comparing dimerization and translocation of TGFβ receptors at the protein level in quail versus duck, which ultimately may provide insight on how intracellular mediators and downstream effectors become differentially expressed among these species.

## Promoter evolution underlies the species-specific expression of Mmp13

To identify regulatory mechanisms underlying species-specific differences in the response of downstream targets to TGFβ signaling, we interrogated the structure and function of the *Mmp13* promoter in chick, quail, and duck. We observe that a 2 kb fragment of the quail and chick *Mmp13* promoter is induced by rTGFβ1, whereas the equivalent promoter region in duck is not. The *Mmp13* promoter contains RUNX2, SMAD, and other binding elements that regulate its expression (*Pendás et al., 1997*; *Selvamurugan et al., 2004a*; *Selvamurugan et al., 2004b*; *Wang et al., 2004*; *Selvamurugan et al., 2006*; *Selvamurugan et al., 2009*; *Chen et al., 2012a*; *Meyer et al., 2016*; *Takahashi et al., 2017*; *Arumugam et al., 2017*; *Young et al., 2019*). Moreover, genetic polymorphisms in the *Mmp13* promoter are known to affect transcriptional activity and phenotypic variation in chick (*Yuan et al., 2016*) and humans (*Ye, 2000*; *Benderdour et al., 2002*; *Yoon et al., 2002*; *Yan and Boyd, 2007*; *Achari et al., 2008*; *Hashimoto et al., 2013*). The SMAD-binding element that we examined has been shown to be regulated by SMAD3 in the human and mouse *Gsc* promoter (*Martin-Malpartida et al., 2017*). Targeting and modifying this SMAD-binding element reduced *Mmp13* transcriptional activity when eliminated from the proximal promoter in quail and chick and increased activity when added to the duck promoter. Thus, this SMAD-binding element is a key transcriptional regulator of *Mmp13* expression that likely accounts for the species-specific sensitivity to TGFβ signaling that distinguishes chick and quail from duck.

Our results also suggest that changes in the levels of *Runx2* may affect *Mmp13* promoter transcriptional activity in a species-specific manner. Quail express higher levels of *Runx2* coincident with their smaller jaws, and we have shown previously that overexpressing *Runx2* significantly reduces jaw size likely by accelerating the timing of osteoblast differentiation and bone deposition (*Hall et al., 2014*). Here, we find that overexpressing *Runx2* can activate the *Mmp13* promoter in quail but not duck. Similarly, other studies have shown that RUNX2 can have both activating and repressive effect on *Mmp13* expression depending on the context and the duration of treatment (*Chen et al., 2012a*). Interestingly, *Runx2* expression is upregulated in quail lower jaws in response to treatments with rTGFβ1, whereas *Runx2* appears to become repressed in duck, but the mechanism underlying this observation is yet to be determined. Taken together, our results suggest that the presence and/or absence of regulatory elements within the *Mmp13* promoter of chick, quail, and duck can facilitate the species-specific response to TGFβ signaling via the differential binding of SMAD and/or RUNX2.

To this point, we identified SNPs directly adjacent to a RUNX2-binding site that distinguish quail and chick (i.e. 'AG') from duck (i.e. 'CA'). We deduce that these SNPs may play an essential role in RUNX2 binding because switching them along with the SMAD-binding element between species abolishes the response to rTGFβ1 with quail and chick promoters but adds a response to the duck

promoter. Altering one of these individual elements alone is not sufficient for mediating a response to rTGFβ1, suggesting that cooperativity at the SMAD-binding element plus the RUNX2-binding element when 'AG' SNPs are present is necessary to achieve induction. Other studies have shown that such cooperative binding and modifications to various sites can affect *Mmp13* promoter activation, including those for methylation of *Hypoxia inducible factor 1 subunit alpha*, as well as binding of *Y-box binding protein-1*, *Lymphoid enhancer binding factor 1*, *Osterix*, *Vitamin D receptor*, and *Parathyroid hormone* (*Samuel et al., 2007*; *Yun and Im, 2007*; *Hashimoto et al., 2013*; *Meyer et al., 2016*). Our findings provide additional evidence that the 'AG' SNPs and SMAD-binding element work cooperatively and are required for TGFβ activation of the *Mmp13* promoter and that species-specific expression levels are mediated by the structure of the promoter. In this regard, evolutionary changes within the promoter of chick and quail versus duck appear to be a central mechanism that controls the species-specific expression of *Mmp13*. Whether these nucleotide differences represent a shared-derived condition for Galliformes (i.e. chick and quail) or for Anseriformes (i.e. duck) remains unclear at this point but could be clarified by examining other avian lineages.

## Multiple hierarchical levels of TGFβ regulation underlie avian jaw evolution

In '*Problems of Relative Growth*,' *Huxley, 1932* proposed genetic mechanisms for generating phenotypic diversity that included mutations affecting what he called time and rate genes. Huxley's critical insight was that these types of mutations could regulate where and when a particular gene turns on or off and titrate its expression, which ultimately would then alter growth parameters and anatomy at many levels in coordinated ways (*Schneider, 2018a*). In this regard, regulatory changes have been viewed as a primary mechanism for evolutionary diversification instead of modifications to coding sequences of genes (*Britten and Davidson, 1969*; *King and Wilson, 1975*; *Carroll, 2005*; *Wray, 2007*; *Romero et al., 2012*). Given that roughly 98% of DNA in the human genome is non-coding (*Clamp et al., 2007*), deciphering the morphogenetic consequences of mutations in regulatory domains is necessary to illuminate fundamental mechanisms of development, disease, and evolution (*Ahituv, 2012*). While many studies have focused on cis-regulatory enhancers, which are often located tens to hundreds of kb from transcriptional start sites (*Heintzman et al., 2007*; *Shen et al., 2012*; *Prescott et al., 2015*; *Schaffner, 2015*; *Long et al., 2016*; *Rebeiz and Tsiantis, 2017*; *Williams et al., 2018*; *Kim et al., 2019*), there are some examples of how evolution within the more proximal promoter itself can drive phenotypic change. The globin genes are a well-studied case (*Pace and Makala, 2012*), although these also appear to rely on distal enhancers for their regulation (*Hay et al., 2016*). When broadly comparing non-coding regulatory regions across species, enhancers appear to undergo rapid evolutionary turnover, whereas promoters tend to remain partially or fully conserved (*Villar et al., 2015*; *Berthelot et al., 2018*). MMP family members are no exception and many share structural similarity and numerous conserved binding elements in their proximal promoters (*Benbow and Brinckerhoff, 1997*; *Samuel et al., 2007*; *Yan and Boyd, 2007*; *Fanjul-Fernández et al., 2010*; *Hashimoto et al., 2013*). For this reason, we think our finding that species-specific changes in the *Mmp13* promoter, which alter its sensitivity to TGFβ signaling and its regulation by RUNX2, offers a novel insight on a potential developmental mechanism for generating evolutionary variation in jaw length. Similarly, polymorphisms in the *Mmp13* promoter that affect binding of transcriptional activators or repressors can generate abnormal variation associated with human disease (*Pendás et al., 1997*; *Ye, 2000*; *Marchenko et al., 2002*; *Yoon et al., 2002*; *Achari et al., 2008*).

While our results indicate that promoter evolution may play an important role in the species-specific expression of *Mmp13* and in this way could influence the establishment of jaw length in quail versus duck via bone resorption, the potential contributions from additional levels of *Mmp13* regulation, multiple *Mmps*, as well as different members and targets of TGFβ and other signaling pathways remain vast (*Mina, 2001b*; *Mina, 2001a*; *Depew et al., 2002*; *Mina et al., 2002*; *Chakraborti et al., 2003*; *Oka et al., 2007*; *Havens et al., 2008*; *Balic et al., 2009*; *Fanjul-Fernández et al., 2010*; *Fish et al., 2011*; *Young et al., 2019*). This may help explain why despite our widespread overexpression of *Mmp13* in NCM, we only observe a 10% foreshortening of the duck lower jaw. Likewise, we predict that species-specific variation in cis-regulatory domains at more distal enhancers, in the complement of available transcriptional cofactors, in epigenetic mechanisms of transcriptional and post-transcriptional control such as DNA methylation and non-coding RNA, in the post-translational

modifications and interactions of proteins, and in the gradients and thresholds of secreted molecules will similarly affect jaw length in some meaningful manner (*Schneider, 2007*; *Schneider, 2015*; *Schneider, 2018b*; *Schneider, 2018a*). Finally, another level of regulation to consider that distinguishes jaw development in quail from duck involves species-specific differences in the cell biological properties of NCM and in the physical and signaling interactions between NCM and adjacent tissues (*Schneider and Helms, 2003*; *Eames and Schneider, 2008*; *Merrill et al., 2008*; *Tokita and Schneider, 2009*; *Solem et al., 2011*; *Fish and Schneider, 2014b*; *Fish et al., 2014c*; *Hall et al., 2014*; *Ealba et al., 2015*; *Woronowicz et al., 2018*; *Woronowicz and Schneider, 2019*). Thus, jaw patterning relies upon a myriad of molecular and cellular mechanisms.

Based on the results of our study, we conclude that differential regulation of the TGFβ pathway at multiple hierarchical levels has served as a key mechanism of evolution in the avian jaw skeleton. Such a conclusion is also supported by the jaw length variation observed in mice and humans following mutations in *Mmp13* and *Runx2* (*Ito et al., 2003*; *Dudas et al., 2006*). Similarly, mutations in *Tgfβ2* and *Tgfβr1* cause microretrognathia in Loeys-Dietz syndrome (*Loeys et al., 2005*; *Zhao et al., 2008*), mutations in *Tgfβr2* and *Smad2* cause jaw length defects (*Nomura and Li, 1998*; *Oka et al., 2007*; *Oka et al., 2008*), and TGFβ signaling has been shown to be essential for patterning the proximal portion of the murine dentary (*Anthwal et al., 2008*). Moreover, while in the current study we have only detailed events during the development of the lower jaw skeleton, there is considerable evidence that patterned outgrowth of the upper jaw also depends upon TGFβ signaling. Examples include the process of palatogenesis where TGFβ signaling and the regulation of MMPs are critical for shelf closure (*Brunet et al., 1995*; *Proetzel et al., 1995*; *Chai et al., 1997*; *Kaartinen et al., 1997*; *Blavier et al., 2001*; *Ito et al., 2003*; *Dudas et al., 2006*; *Iwata et al., 2011*), the pathogenesis of Marfan syndrome where elevated TGFβ signaling causes excessive upper jaw growth (*Westling et al., 1998*; *Neptune et al., 2003*), and the dysregulation of non-canonical NCM-mediated TGFβ signaling, which results in hypoplastic facial features (*Yumoto et al., 2013*). But exactly how such alterations to members and targets of the TGFβ pathway can ultimately modulate the complex features of jaw morphology requires further elucidation. While the TGFβ signaling axis has been deeply conserved across vertebrates, our study provides insight into ways this pathway can evolve over time, become differentially regulated, and serve as a source of phenotypic variation. By focusing on species-specific regulation of the TGFβ pathway in the jaws of anatomically distinct birds, we hope our work has helped pinpoint precisely when, where, and how one mode of change in a developmental program can alter the course of evolution.

## Materials and methods
### The use of avian embryos
Fertilized eggs of chicken (*Gallus gallus*), Japanese quail (*Coturnix coturnix japonica*), and white Pekin duck (*Anas platyrhynchos domestica*) were purchased commercially (AA Lab Eggs, Westminster, CA) and incubated at 37.8°C in a humidified chamber (GQF Hova-Bator 1588, Savannah, GA) until they reached embryonic stages appropriate for analyses (see *Supplementary file 1* for details on materials, reagents, equipment, supplies, and software used in this study). For all experiments, we adhered to accepted practices for the humane treatment of avian embryos as described in S3.4.4 of the AVMA Guidelines for the Euthanasia of Animals: 2013 Edition (*Leary et al., 2013*). Embryos were matched at equivalent stages using the Hamburger and Hamilton (HH) staging system, a well-established standard that is based on external morphological characters, is independent of body size and incubation time (*Hamburger and Hamilton, 1951*; *Hamilton, 1965*) and can be adapted to other avian species such as quail and duck (*Ricklefs and Starck, 1998*; *Starck and Ricklefs, 1998*; *Schneider and Helms, 2003*; *Lwigale and Schneider, 2008*; *Jheon and Schneider, 2009*; *Ainsworth et al., 2010*; *Mitgutsch et al., 2011*; *Fish and Schneider, 2014a*; *Smith et al., 2015*).

### Histological staining and immunohistochemistry
Chick, quail, and duck embryos were collected at HH40 and fixed in 4% paraformaldehyde (PFA; 15714, Electron Microscopy Sciences, Hatfield, PA) overnight at 4°C (*Schneider, 1999*; *Schneider et al., 2001*). To detect TRAP in whole mount, chick, quail, and duck lower jaws were skinned and stained for 1.5 hr at 37°C using the Acid Phosphatase Leukocyte kit (387A-1KT, MilliporeSigma,

Burlington, MA) following the manufacturer's protocol, except 7 mg/ml Fast Red Violet (F3381, MilliporeSigma, Burlington, MA) was used in place of the Fast Garnet GBC Base solution (*Ealba et al., 2015*). Samples were cleared in 100% glycerol (525342 C, Thermo Fisher Scientific, Waltham, MA). For sections, embryos were dehydrated in methanol, embedded in paraffin, and cut into 10 µm sagittal sections. Sections were deparaffinized, rehydrated, and adjacent sections were stained with Milligan's trichrome at room temperature to detect bone deposition (*Presnell and Schreibman, 1997*) or stained with the Acid Phosphatase Leukocyte kit at 37°C for 1 hr to detect bone resorption (*Ealba et al., 2015*).

IHC was performed on adjacent sections. For antigen retrieval, sections were heated in a microwave to 95°C in 10 mM sodium citrate buffer for 10 min, and endogenous peroxidase activity was blocked with 3% hydrogen peroxide for 15 min. Sections were incubated with 1 µg/ml of a custom-made MMP13 rabbit polyclonal primary antibody (GenScript, Piscataway, NJ; *Supplementary file 2*) overnight at 4°C. Sections were labeled with 1:500 goat anti-rabbit Alexa Fluor 647 secondary antibody (A32733, Thermo Fisher Scientific, Waltham, MA) overnight at 4°C. 10 mg/ml Hoechst 33,342 dye (62249, Thermo Fisher Scientific, Waltham, MA) was diluted 1:100 and used to stain nuclei. Lower jaw sections were imaged using a Nikon AZ100 C2 macroconfocal microscope and image acquisition system (Nikon Instrument, Inc, Melville, NY) for IHC, and a Leica DM 2500 (Leica Microsystems, Inc Buffalo Grove, IL) with a color digital camera system (SPOT Insight 4 Megapixel CCD, Diagnostic Instruments, Inc, Sterling Heights, MI) for trichrome and TRAP.

To visualize lower jaw morphology in whole mount, chick, quail, and duck heads were fixed in 4% PFA overnight and stained for 20 min with 0.02% ethidium bromide (1610433, Bio-Rad, Hercules, CA) using a previously published protocol (*Eames and Schneider, 2005*). Samples were washed three times in 1× PBS. Ethidium bromide-stained and TRAP-stained samples were imaged on a dissecting microscope (MZFLIII, Leica Microsystems, Inc, Buffalo Grove, IL) using either epifluorescent, transmitted, and/or incident illumination and a color digital camera system (SPOT Insight 4 MP).

## RNAscope in situ hybridization assay

Custom RNAscope probes (Advanced Cell Diagnostics, Inc, Newark, CA) for in situ hybridization were designed using species-specific sequences for *Tgfβ1* and *Tgfβr1* derived from National Center for Biotechnology Information (NCBI) databases as well as a bulk RNA-seq dataset for the lower jaws of chick, quail, and duck at HH37. Ubiquitin C and peptidyl-prolyl cis-trans isomerase B were used as positive controls. In situ hybridization was performed on near adjacent sections to those used for trichrome, TRAP, and IHC staining on chick, quail, and duck HH40 lower jaws. Sections were deparaffinized and in situ hybridization was performed following the RNAscope Multiplex Fluorescent Reagent Kit v2 Assay protocol (Document #323100-USM, Advanced Cell Diagnostics, Inc, Newark, CA). Briefly, sections were treated with hydrogen peroxide solution for 10 min at room temperature. Slides were washed and target retrieval was performed using a steamer for 15 min in RNAscope Target Retrieval Reagent. Samples were incubated with RNAscope Protease Plus Reagent and incubated in a HybEZ II Oven at 40°C for 20 min. Tissue sections were hybridized with probes for 2 hr in the HybEZ II Oven at 40°C. Slides were then hybridized in BaseScope v2 AMP 1 for 30 min, BaseScope v2 AMP 2 for 30 min, and BaseScope v2 AMP 3 for 30 min, all at 40°C. Signal was developed in RNAScope Multiplex FL v2 HRP-C1 for 15 min at 40°C. Opal dyes 570 and 690 were utilized as fluorophores and incubated at 40°C for 30 min. Samples were counterstained with 4',6-diamidino-2-phenylindole (DAPI).

Sections were imaged on a confocal microscope (SP8, Leica Microsystems, Inc, Buffalo Grove, IL), equipped with an HC PL APO 20×/0.75 IMM CORR CS2 lens. DAPI was imaged using a 405 nm laser with an emission band pass of 435–475 nm using a photomultiplier tube detector. Opal 570 was imaged by setting a white light laser to 550 nm with an emission band pass of 560–585 nm using a hybrid detector. Opal 690 was imaged by setting a white light laser to 670 nm with an emission band pass of 585–730 nm using a hybrid detector.

## RNA extraction

Lower jaws were dissected from chick, quail, and duck embryos at HH31, HH34, HH37, and HH40, and total RNA was extracted using the RNeasy Plus Mini Kit (74136, Qiagen, Hilden, Germany) following the manufacturer's protocol. Lower jaws were resuspended in 600 µl of RTL plus buffer supplemented with 1% β-mercaptoethanol (M3148-100ML, MilliporeSigma, Burlington, MA) and Reagent DX (19088,

Qiagen, Hilden, Germany). HH31 and HH34 lower jaws were processed in a Bead Mill 24 Homogenizer (15-340-163, Fisher Scientific, Waltham, MA) at 5 m/s for 30 s with 1.4 mm ceramic beads (15-340-153, Fisher Scientific, Waltham, MA). HH37 and older lower jaws were homogenized at 5 m/s for 60 s with 2.8 mm ceramic beads (15-340-154, Fisher Scientific, Waltham, MA). Following purification of total RNA, residual genomic DNA was removed using TURBO DNA-free Kit (AM1907, Invitrogen, Carlsbad, CA). DNased RNA was reverse-transcribed using iSCRIPT (1708841, Bio-Rad).

## Quantitative PCR

Gene expression was analyzed by qPCR with iQ SYBR Green Supermix (1708882, Bio-Rad, Hercules, CA) and normalized to 18S rRNA following previously published protocols (*Dole et al., 2015*; *Smith et al., 2016*). Primers were designed (Geneious Prime, Version 2020.2.4) to amplify conserved regions among chick, quail, and duck for members and targets of the TGFβ pathway including *Tgfβ1*, *Tgfβ2*, *Tgfβ3*, *Tgfβr1*, *Tgfβr2*, *Tgfβr3*, *Acvrl1*, *Smad2*, *Smad3*, *Runx2*, *Mmp2*, *Mmp9*, *Mmp13*, *Mmp14*, *Pai1*, *Col1a1*, *Ocn*, *Ctsk*, and *Sost* (*Supplementary file 3*). Criteria for experimental design included limiting primers to 20 bp in length, amplifying regions of ~150 bp, using an annealing temperature of 60°C, keeping GC content around 50%, minimizing self-complementarity (i.e. primer-dimers), and amplifying regions that span exon-exon junctions. To account for alternative efficiencies of primer binding between species, data were normalized using serial dilutions of pooled cDNA and a standard curve method (*Ealba and Schneider, 2013*; *Dole et al., 2015*; *Smith et al., 2016*). Each sample was assayed in technical duplicate.

## Western blots

Lower jaws were lysed with 1× radioimmunoprecipitation assay (RIPA) lysis buffer (20–188, MilliporeSigma, Burlington, MA) containing Halt protease inhibitors (78430, Thermo Fisher Scientific, Waltham, MA). A BCA assay (23225, Thermo Fisher Scientific, Waltham, MA) was performed to quantify protein using a SpectraMax M5 microplate reader (Molecular Devices, San Jose, CA). 40 µg of protein was electrophoresed on a 10% sodium dodecyl sulfate (SDS) polyacrylamide gel as previously described (*Smith et al., 2016*). Proteins were transferred to an Immobilon-P PVDF membrane (IPVH00010, MilliporeSigma, Burlington, MA). Membranes were probed with 1:1000 rabbit anti-human pSer423/pSer425 SMAD3 antibody (NBP1-77836, Novus Biologicals, Littleton, CO), 1:1000 rabbit anti-human SMAD3 antibody (NB100-56479, Novus Biologicals, Littleton, CO), 1 µg/ml rabbit anti-chick MMP13 custom-made primary antibody (GenScript, Piscataway, NJ; *Supplementary file 2*), 1:1000 rabbit anti-human MMP2 antibody (NB200-193, Novus Biologicals, Littleton, CO), 1:4000 mouse anti-human β-actin antibody (NB600-501, Novus Biologicals, Littleton, CO), 1:15,000 goat anti-rabbit IRDye 800CW (925–32211, LI-COR, Lincoln, NE), and 1:15,000 donkey anti-mouse IRDye 680RD antibody (*Supplementary files 1 and 2*; 925–68072, LI-COR, Lincoln, NE). Fluorescent signal was detected using the Odyssey Imaging System (LI-COR, Lincoln, NE). Quantifications of protein bands were performed using Image Studio Lite. Protein levels were normalized to β-actin.

## RNA-seq, data alignment, and normalization

cDNA from HH37 chick, quail, and duck lower jaws were synthesized using an NEB Next Single Cell/Low Input cDNA Synthesis Amplification Module (E6421, New England BioLabs, Ipswich, MA) following PacBio Iso-Seq template preparation guidelines. cDNA was barcoded using the PCR barcoding expansion 1–96 kit (EXP-PBC096, Oxford Nanopore Technologies, Oxford, UK). cDNA samples were pooled, and a sequencing library was prepared using the ligation sequencing kit (SQK-LSK110, Oxford Nanopore Technologies, Oxford, UK). 50 fmol of the final library was loaded onto each of the two PromethION flowcells (v R9.4.1), and the run was performed for 72 hr. Basecalling and de-multiplexing were performed live on the PromethION compute module (Basecaller Version ont-Guppy-for-minKNOW 4.0.11).

To calculate and normalize read counts for TGFβ ligands and receptors (i.e. *Tgfβ1*, *Tgfβ2*, *Tgfβ3*, *Tgfβr1*, *Tgfβr2*, and *Tgfβr3*), as well as *Mmp13*, mRNA sequences for chick, quail, and duck were aligned using Minimap2 (*Li, 2018*) with the Nanopore preset in Geneious Prime set to chick, quail, and duck coding sequences extracted from NCBI. Library normalization factors were calculated by inputting the read counts for each gene into the ruvSEQ (*Risso et al., 2014*) and edgeR Bioconductor packages (*Gentleman et al., 2004*; *Robinson et al., 2010*).

## Culture experiments and TGFβ pathway manipulation

For in vitro experiments, an embryonic chick fibroblast cell line (DF-1, CRL-12203, ATCC, Manassas, VA) and an embryonic duck fibroblast cell line (CCL-141, ATCC, Manassas, VA) were cultured in complete media (Dulbecco's Modified Eagle's Medium [DMEM], 10–013-CV, Corning, Corning, NY) for chick fibroblasts or MEMα (Thermo Fisher Scientific, Waltham, MA, A10490-01) for duck fibroblasts supplemented with 10% fetal bovine serum (FBS, 97068–085, Lot# 283K18, VWR, Radnor, PA) and 1× penicillin-streptomycin (15140122, Thermo Fisher Scientific, Waltham, MA). Cells were screened monthly for mycoplasma contamination. Cells were plated; serum-deprived for 12 hr; treated with 5 ng/ml recombinant (r) human TGFβ1 derived from HEK293 cells (100–21, PeproTech, Rocky Hill, NJ) for 1, 3, 6, or 24 hr; and then harvested for mRNA and protein analysis. For luciferase assays, cells were treated with 5 ng/ml rTGFβ1 for 24 hr. TGFβ signaling was disrupted by treating cells with 1 µM SB431542 (S4317, MilliporeSigma, Burlington, MA), which inhibits TGFβR1 activity (*Inman et al., 2002*; *Laping et al., 2002*; *Vogt et al., 2011*), or with 3 µM SIS3 (5291, Tocris/R&D Systems, Minneapolis, MN) which inhibits SMAD3 (*Jinnin et al., 2006*). SB431542 and SIS3 were solubilized in 50 mM dimethyl sulfoxide (DMSO) and delivered alone or in combination with rTGFβ1 for 24 hr.

For ex vivo experiments, HH34 lower jaws from chick, quail, and duck were dissected, placed on a 0.45 µm membrane filter (HAWP01300, MilliporeSigma, Burlington, MA), cultured for 24 hr in six-well transwell inserts (10769–192, VWR, Radnor, PA) in complete media (i.e. DMEM), switched to media supplemented with 1% FBS and 1× penicillin-streptomycin, and treated with 10, 25, or 50 ng/ml rTGFβ1 in the media for 6 or 24 hr. Lower jaws were then collected for mRNA and protein analyses. To disrupt the TGFβ pathway, lower jaws from quail and duck were harvested at HH35. Affigel Blue Beads (1537301, 250–300 µm diameter, 50–100 mesh, Bio-Rad, Hercules, CA) were washed with PBS and soaked in either 10 mM SB431542, 10 mM SIS3, 1 mg/ml MMP13 inhibitor (444283, MilliporeSigma, Burlington, MA), 160 µg/ml rTGFβ1, or 100 µg/ml rMMP13 (4442875, MilliporeSigma, Burlington, MA) for 1 hr at room temperature. Concentrations were based on those used previously (*Ealba et al., 2015*; *Havis et al., 2016*; *Woronowicz et al., 2018*). Controls beads were soaked in 10% DMSO for SB431542, SIS3, and MMP13 inhibitor treatments. Control and treatment beads were surgically inserted into the left and right sides (respectively) using forceps, and lower jaws were placed on a 0.45 µm membrane filter, put in transwell inserts, cultured in complete media (i.e. DMEM) supplemented with 50 µg/ml ascorbic acid (A61-25, Thermo Fisher Scientific, Waltham, MA) and 10 mM β-glycerol phosphate (AC410991000, Thermo Fisher Scientific, Waltham, MA) for 5 days, and collected for whole mount TRAP staining.

## Quantification of TRAP staining

To quantify TRAP staining, images of lower jaws treated with TGFβR1, SMAD3, or MMP13 inhibitors were adjusted in Adobe Photoshop 2022 (Version 23.2.2) to normalize for exposure, brightness, contrast, saturation, and color balance across samples. Exclusion criteria comprised samples where control and/or treatment beads had fallen out or were misplaced, samples that became substantially malformed or stunted during culture, and samples that were uniformly over- or understained with TRAP. The Rectangular Marquee tool was used in Photoshop to define a 1 mm square area (200 × 200 pixels) centered around either the control or treatment bead on each side of the lower jaw. Images were cropped to the 1 mm square. The Elliptical Marquee tool was used to delete an equally sized area that covered the beads in each pair of cropped images (i.e. control versus treated sides of the same sample). Cropped images were opened in ImageJ (Fiji Version 2.1.0/1.53 g; *Schindelin et al., 2012*; *Schneider et al., 2012*). Cropped images were adjusted using the Color Threshold tool and the default method so that the same thresholding value was applied to the control and treated sides of each pair. Thresholded images were analyzed using the analyze particles function and results were saved as percent (%) area to represent the total amount of TRAP-positive staining (*Sawyer et al., 2003*; *Holland et al., 2019*; *Mira-Pascual et al., 2020*).

## Generation of Mmp13 and Runx2 overexpression constructs

To generate *Mmp13 and Runx2* overexpression constructs, full-length cDNA was synthesized using Maxima H Minus first strand cDNA synthesis kit (K1651, Thermo Fisher Scientific, Waltham, MA) following the manufacturer's protocol with 2 µg of total HH37 chick, quail, or duck lower jaw RNA and 100 pmol of d(T)20 VN primer (*Chu et al., 2020*). The cDNA synthesis reaction was carried out at

50°C for 30 min, 55°C for 10 min, 60°C for 10 min, 65 °C for 10 min, and 85 °C for 5 min. Full-length *Mmp13* and *Runx2* were amplified by PCR using Q5 Hot Start High-Fidelity DNA polymerase and cloned using CloneJET PCR Cloning Kit. Full-length *Mmp13* and *Runx2* were confirmed by Sanger sequencing and cloned into our pPIDNB custom-made plasmid (*Chu et al., 2020*), which was digested with AflII (R0520S, NEB, Ipswich, MA) and PstI (R3140S, NEB, Ipswich, MA), using NEBuilder HiFi DNA Assembly Master Mix. The pPIDNB plasmid contains a constitutively active mNeongreen fluorescent protein (GFP) (*Shaner et al., 2013*), which serves as a reporter for transfection or electroporation efficiency; and a dox-inducible (*Gossen et al., 1995*; *Loew et al., 2010*; *Heinz et al., 2011*) mScarlet-I red fluorescent protein (RFP) (*Bindels et al., 2017*). Constructs were verified by sequencing and midi-prepped for transfection or electroporation using PureLink Fast Low-Endotoxin Midi Kit (A35892, ThermoFisher Scientific, Waltham, MA).

Enzymatic activity of overexpressed MMP13 protein was validated by transfecting HEK293 cells with pPIDNB-*Mmp13* or empty pPIDNB vector using Lipofectamine 3000 (L3000008, Invitrogen, Carlsbad, CA). Cells were recovered after 16 hr with DMEM supplemented with 10% FBS, 1× penicillin-streptomycin and 100 ng/ml dox. After 4 days, the cell culture medium was harvested. MMP13 was activated by treating the cell culture medium with 1 mM 4-aminophenylmercuric acetate (A9563-25G, MilliporeSigma, Burlington, MA, USA) for 2 hr at 37°C. Collagenase activity of the activated cell culture medium was assayed using an EnzChek Gelatinase/Collagenase Assay Kit (E-12055, Thermo Fisher Scientific, Waltham, MA) according to the manufacturer's protocol with 3 µl of gelatin per 200 µl reaction. Reactions were carried out in black 96-well plates (655079, Greiner Bio-One, Monroe, North Carolina, USA). Fluorescence was measured using an iD5 plate reader microplate reader (Molecular Devices, San Jose, CA, USA). Raw arbitrary fluorescence units were normalized to the control (pPIDNB empty vector) and represented as relative fluorescence units (*Figure 5—figure supplement 1A*).

## In ovo electroporation

In ovo electroporations were performed as described previously (*Chu et al., 2020*). Briefly, approximately 0.4 µl of a solution of Fast Green dye plus duck pPIDNB-*Mmp13* at 3 µg/µl and pNano-hyPBase at 1 µg/µl were mixed, and approximately, 0.05 µl was injected with a Pneumatic PicoPump (PV830, World Precision Instruments, Sarasota, FL) into HH8.5 duck anterior neural tubes using thin wall borosilicate glass micropipettes (O.D. 1.0 mm, I.D. 0.75 mm, B100-75-10, Sutter Instrument, Novato, CA) pulled on a micropipette puller (P-97 Flaming/Brown, Sutter Instrument, Novato, CA). Homemade platinum electrodes (78–0085, Strem Chemicals Inc, Fisher Scientific, Hanover Park, IL) mounted in an Adjustatrode Holder (01-925-09, Intracel by Abbotsbury Engineering Ltd., St Ives, UK) were positioned on each side of the area pellucida and centered at the midbrain-hindbrain boundary and along the neural folds. The distance between electrodes was set to 5 mm. The electrodes were overlayed with albumin to prevent drying and to facilitate conductivity. Three square pulses (1 ms long, 50 volt, with 50 ms spaces), followed by five square pulses (50 ms long, 10 volt, with 50 ms spaces), were administered (CUY21EDITII Next Generation Electroporator, BEX CO, Ltd, Tokyo, Japan) as done previously to allow unilateral entry of DNA into the presumptive NCM destined for the mandibular arch (*Creuzet et al., 2002*; *Krull, 2004*; *McLennan and Kulesa, 2007*; *Hall et al., 2014*). The contralateral (un-electroporated) side served as an internal control. After electroporation, a small amount of albumin was added on top of the embryo to prevent desiccation. Eggs were sealed with tape and re-incubated.

Embryos were allowed to develop until HH35 and then were treated in ovo with a single dose of 3.75 µg of doxycycline hyclate (446060250, Acros Organics, Geel, Belgium) in 750 µl of Hanks' Balanced Salt Solution (14175095, ThermoFisher Scientific by Life Technologies Corporation, Grand Island, NY). Eggs were sealed with tape and re-incubated. Treated embryos were collected at HH40, placed in 4% PFA overnight at 4°C, washed in 1× PBS, and dehydrated in 70% ethanol (EtOH). Electroporation efficiency and extent of overexpression were evaluated at the time of embryo collection by detecting GFP and RFP (*Chu et al., 2020*) on either a stereodissecting microscope (MZFLIII, Leica Microsystems, Inc, Buffalo Grove, IL) under epifluorescent illumination or on a macro confocal microscope (Nikon AZ100 C2, Nikon Instrument, Inc, Melville, NY).

## Microcomputed tomography and morphometrics

Duck heads were placed in 50 ml Falcon tubes and scanned using a SCANCO Medical μCT 50 cabinet cone-beam μCT at a resolution of 10 μm. All specimens were scanned using the same energy/intensity settings (55 kVp, 109 μA, 6 W), calibration settings (55 kVp, 0.5 mm Al filter, beam hardening: 1200 mg HA/ccm), and a 0.5 mm Al filter (Skeletal Biology and Biomechanics Core, Core Center for Musculo-skeletal Biology and Medicine, UCSF). Scans were reconstructed in the SCANCO Medical software scan process and saved as DICOM files. 3D meshes of each specimen were created using Dragonfly software (Version 4.1.0.647, Object Research Systems, Montreal, Canada). Briefly, the DICOM files were imported and segmented using the image stack histogram to determine an equivalent segmentation threshold for all samples. The resulting region of interest was closed then smoothed (morphological operations, both with a 3D spherical 7 px kernel size) before being exported to a 'normal' 3D mesh which was subsequently smoothed five times. This provided a 3D model that could then be annotated with landmarks using the points tool, producing 3D coordinates from which lower jaw distances were calculated.

A total of 15 landmarks were annotated on the surface of the lower jaw for each 3D model (*Figure 5—figure supplement 1B,C*). The most distal points of each side of the jaw were landmarked, as well as the most proximal point, and six points were spaced equidistantly along each side of the jaw between the most proximal and most distal point. Using the X, Y, and Z coordinate data, the distance between the most proximal and most distal point of each side of the jaw was calculated as the sum of the 3D vector lengths between pairs of points on each jaw side.

## Mmp13 promoter sequencing

*Mmp13* promoters for chick, quail, and duck were sequenced through inverse PCR, which enables amplification of unknown sequences that flank a known region (*Ochman et al., 1988*; *Green and Sambrook, 2019*). Genomic DNA for inverse PCR was extracted from embryonic chick, quail, and duck tissues using the Purelink Genomic DNA mini kit (K1820-01, Invitrogen, Carlsbad, CA) following the manufacturer's protocol. The sequences for exon 1 of chick, quail, and duck were used as the anchor for designing primers and determining restriction sites. Genomic DNA for the inverse PCR was digested with EcoRI-HF (R3101S, NEB, Ipswich, MA). Digested genomic DNA was purified with GeneJET PCR Purification Kit (K0702, Thermo Fisher Scientific, Waltham, MA, USA) and then ligated with Rapid DNA Ligation Kit (K1422, Thermo Fisher Scientific, Waltham, MA). Inverse PCR was performed on the ligated genomic DNA using Q5 Hot Start High-Fidelity DNA Polymerase (M0493L, NEB, Ipswich, MA). PCR products underwent primer walking Sanger sequencing (*Sterky and Lundeberg, 2000*). For chick, the EcoRI inverse PCR yielded a desired length of 2 kb of promoter sequence, but for duck and quail, less than 2 kb of the promoter was initially sequenced, so inverse PCR was repeated using XbaI (R0145S, NEB, Ipswich, MA).

## Mmp13 promoter sequence analysis

To identify transcription factor-binding sites, we used the JASPAR 2020 database, which contains transcription factor-binding profiles stored as position frequency matrices (*Fornes et al., 2019*). To map transcription factor-binding sites onto the *Mmp13* promoter sequences of chick, quail, and duck, we used the TFBSTools (*Tan and Lenhard, 2016*) R/bioconductor package (*R Development Core Team, 2013*). For the proximal region of the *Mmp13* promoter (i.e. –184 bp for chick and quail, and –181 bp for duck), all vertebrate transcription factors were included in the analysis. For the –2 kb promoter region, only RUNX2 (ID = MA0511), SMAD3_1 (ID = PB0060), SMAD3_2 (ID = PB0164), SMAD2-SMAD3-SMAD4 (ID = MA0513), SMAD3 (ID = MA0795), SMAD4 (ID = MA1153), and SMAD2/3 (ID = MA1622) were included in the analysis since these are TGFβ activated (*Heldin et al., 1997*; *Derynck et al., 1998*; *Derynck et al., 2008*). Position-frequency matrices were converted to position-weighted matrices by setting pseudocounts to 0.8 (*Nishida et al., 2009*) and background frequencies of nucleotides to 0.25. The minimum threshold score was set to 95% for the –184/181 bp promoter region and 90% for the –2 kb promoter region.

A SMAD-binding element (i.e. 5'-GGC(GC/CG)–3'), which was not annotated in the JASPAR 2020 database, was manually added (*Martin-Malpartida et al., 2017*). Sequence logos were generated using the seqLogo function in TFBSTools (*Figure 6—figure supplement 1*).

## Generation of Mmp13 promoter constructs

Each *Mmp13* promoter sequence was amplified by PCR using Q5 Hot Start High-Fidelity DNA Polymerase (M0493L, NEB, Ipswich, MA). To generate luciferase constructs, pGL3 was digested with HindIII-HF (R3104S, NEB, Ipswich, MA) and XhoI (R0146S, NEB, Ipswich, MA). The amplified *Mmp13* promoter sequences and digested pGL3 were purified using GeneJET PCR Purification Kit and cloned using NEBuilder HiFi DNA Assembly Master Mix (E2621L, NEB, Ipswich, MA). Mutations in *Mmp13* promoter SNPs were generated through site-specific mutagenesis PCR with the mutations in the primers (*Ho et al., 1989*). All constructs were verified by sequencing and midi-prepped for transfection using PureLink Fast Low-Endotoxin Midi Kit (A36227, Invitrogen, Carlsbad, CA).

## Transfection and luciferase assay

Cells were plated at 65,000 cells/cm$^2$ in 24-well plates (353047, Corning, Corning, NY). Cells were transfected in each well using 1.5 µl Lipofectamine 3000 (L3000008, Invitrogen, Carlsbad, CA), 1.5 µl P3000 reagent, 150 ng of β-galactosidase transfection efficiency control construct, and 1500 ng of *Mmp13* promoter luciferase construct, or 750 ng of *Mmp13* promoter luciferase construct when combined with 400 ng of pPIDNB-*Runx2* overexpression construct. Cells were transfected for 18 hr, recovered in complete media conditions for 8 hr, and then serum deprived in DMEM for DF-1 cells and MEMα for CCL-141 cells without FBS for 18 hr. Cells transfected with pPIDNB-*Runx2* were treated with a final concentration of 100 ng/ml of dox (446060250, Acros Organics, Fair Lawn, NJ) in DMEM or MEMα without FBS for 24 hr. Cells were lysed in 1× lysis buffer (E1531, Promega, Madison, WI) and analyzed for luciferase activity using beetle luciferin (E1602, Promega, Madison, WI) and coenzyme A (J13787MF, ThermoFisher Scientific, Waltham, MA) normalized to β-galactosidase activity using Galacto-Star β-Galactosidase Reporter (T1012, Invitrogen, Carlsbad, CA) as previously described (*Chen et al., 2012b*). Luminescence was measured using a SpectraMax M5 luminometer. At least two preparations of each DNA construct were tested for the overexpression experiments.

## Electrophoretic mobility shift assay

To test for potential SMAD and RUNX2 protein-binding interactions with the *Mmp13* promoter, a biotin labeled tag was added to synthesized oligonucleotides containing either chick/quail or duck *Mmp13* promoter sequence. To test for potential SMAD protein-binding interactions with the *Mmp13* promoter, we utilized recombinant human SMAD4 protein (ab81764, Abcam, Cambridge, UK). To test for potential RUNX2 protein-binding interactions with the *Mmp13* promoter, chick cells were transfected with quail or duck pPIDNB-*Runx2* and treated with a final concentration of 100 ng/ml of dox after 24 hr. Cells were processed via a nuclear extraction kit (ab113474, Abcam, Cambridge, UK) following the manufacturer's protocol. SMAD4 protein or nuclear extract and *Mmp13* promoter oligos were run using the LightShift Chemiluminescent EMSA Kit (20148, ThermoFisher Scientific). The binding reactions with purified protein or cell extract incorporated binding buffer, 50 ng/µl Poly (dI.dC), 2.5% glycerol, 0.05% NP-40, 50 mM KCl, 5 mM MgCl$_2$, 10 mM ethylenediaminetetraacetic acid (EDTA), and 100 ng of DNA and were incubated for 20 min at room temperature. Samples were run on a 6% polyacrylamide gel in a 0.5× buffer solution containing a mixture of Tris base, boric acid, and EDTA (TBE) and then transferred to a positively charged nylon membrane. The membrane was crosslinked for 10 min using a UV transilluminator. The membrane was incubated with a stabilized streptavidin-horseradish peroxidase conjugate for 15 min, washed, and incubated with luminol/enhancer and peroxide solution. For controls, we used competitor oligos composed of the same *Mmp13* promoter sequences without biotinylation. Each treatment group was incubated in a 10× excess of competitor oligos. Membranes were imaged utilizing chemiluminescence on an ImageQuant LAS 4000.

## Statistics and image processing

Statistical analyses and graphing of data were performed using Prism (Version 9.3.1, GraphPad Software). Data are represented as a mean and error bars represent ± standard error of the mean (SEM). Statistical significance was determined through two-tailed ANOVA adjusted for multiple comparisons using the Bonferroni method for all experimental data, except for RNA-seq comparisons or comparisons between control and treatment groups for TRAP quantification, for which a paired Student's t-test was used. For in ovo data, n refers to the total number of embryos analyzed per group. For in

vitro data, n refers to the total number of individual wells analyzed per group, with each experiment replicated at least three times. For qPCR and luciferase experiments, each sample was run in technical duplicate and averaged. If outliers were found within data sets, a standard Q-test was performed with no more than one outlier removed from any group. In all figures, $p \leq 0.05$ was considered statistically significant, although some statistical comparisons reached significance below $p \leq 0.01$, $p \leq 0.001$, or $p \leq 0.0001$ as noted. Group size 'n' is denoted in the figure legends. Formal power analyses were not conducted. Images were were adjusted in Adobe Photoshop 2022 (Version 23.2.2) to normalize for exposure, brightness, contrast, saturation, and color balance across samples. Figures were assembled in Adobe Illustrator 2022 (Version 26.2.1).

## Acknowledgements

We thank T Alliston and R Marcucio for helpful discussions; Z Vavrušová, A Nguyen, A Lucena, G Krish, P Asfour, and T Huang for technical assistance. We thank T Dam at AA Lab Eggs. The pmScarlet-i_C1 was a gift from Dorus Gadella (Addgene, #85044). The AAVS1 Puro Tet3G 3xFLAG Twin Strep was a gift from Yannick Doyon (Addgene, # 92099). The pKanCMV-mClover3- mRuby3 was a gift from Michael Lin (Addgene, #74252). The pCAG-Cre-IRES2-GFP was a gift from Anjen Chenn (Addgene, #26646). The pCMV-hyPBase was provided by the Wellcome Trust Sanger Institute. The mNeon-Green was provided by Allele Biotechnology & Pharmaceuticals. This work was supported in part by the UCSF Biological Imaging Development Core (BIDC); the UCSF Core Center for Musculoskeletal Biology and Medicine (CCMBM) through NIAMS P30 AR066262; NIDCR F31 DE027283 to S.S.S.; and NIDCR R01 DE016402 and R01 DE025668, and NIH Office of the Director S10 OD021664 to R.A.S.

## Additional information

### Funding

| Funder | Grant reference number | Author |
| --- | --- | --- |
| National Institutes of Health | R01 DE016402 | Richard A Schneider |
| National Institutes of Health | R01 DE025668 | Richard A Schneider |
| National Institutes of Health | S10 OD021664 | Richard A Schneider |
| National Institutes of Health | F31 DE027283 | Spenser S Smith |

The funders had no role in study design, data collection and interpretation, or the decision to submit the work for publication. For the purpose of Open Access, the authors have applied a CC BY public copyright license to any Author Accepted Manuscript version arising from this submission.

### Author contributions

Spenser S Smith, Data curation, Formal analysis, Funding acquisition, Investigation, Methodology, Supervision, Validation, Visualization, Writing - original draft, Writing – review and editing; Daniel Chu, Data curation, Formal analysis, Investigation, Methodology, Validation, Visualization, Writing – review and editing; Tiange Qu, Formal analysis, Investigation, Validation, Visualization, Writing – review and editing; Jessye A Aggleton, Formal analysis, Investigation, Visualization, Writing – review and editing; Richard A Schneider, Conceptualization, Formal analysis, Funding acquisition, Methodology, Project administration, Resources, Supervision, Visualization, Writing - original draft, Writing – review and editing

### Author ORCIDs

Spenser S Smith http://orcid.org/0000-0002-3984-3174
Daniel Chu http://orcid.org/0000-0003-3697-8003
Richard A Schneider http://orcid.org/0000-0002-2626-3111

### Ethics

For all experiments, we adhered to accepted practices for the humane treatment of avian embryos as described in S3.4.4 of the AVMA Guidelines for the Euthanasia of Animals: 2013 Edition (Leary et al., 2013).

### Decision letter and Author response

Decision letter https://doi.org/10.7554/eLife.66005.sa1
Author response https://doi.org/10.7554/eLife.66005.sa2

---

## Additional files

### Supplementary files

• Supplementary file 1. Materials, reagents, equipment, supplies, and software used in this study.

• Supplementary file 2. MMP13 antigen sequence.

• Supplementary file 3. Primer sequences used for PCR and qPCR analysis.

• Supplementary file 4. Potential binding elements within the Mmp13 promoter predicted via the JASPAR 2020 database.

• Transparent reporting form

### Data availability

Data generated or analysed during this study are included in the manuscript and supporting files. Source data files have been provided for Figures 2, 3, 4, and 5. GenBank accession numbers for nucleotide sequences are as follows: Runx2 (MW036689) and Mmp13 (MW036690). Plasmids are also available at Addgene (https://www.addgene.org/Richard_Schneider/) subject to the terms of the original licenses under which they were obtained. The RNA-seq dataset for chick, quail, and duck mandibular primordia at HH37 has been deposited in Dryad (https://doi.org/10.7272/Q62805W5).

The following dataset was generated:

| Author(s) | Year | Dataset title | Dataset URL | Database and Identifier |
|---|---|---|---|---|
| Smith SS, Chu D, Schneider RA | 2022 | RNA seq data set of chick, quail, and duck mandibular primordia at embryonic stage (HH) 37 | https://doi.org/10.7272/Q62805W5 | Dryad Digital Repository, 10.7272/Q62805W5 |

---

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
