## [Editor Report]

The manuscript brings new original findings about developmental mechanisms regulating MMP13 activity and associated bone resorption in avian species. These processes lead to the control of jaw size in a species-specific context, therefore, indicating probable evolutionary significance.

---

## [Decision Letter]

**Decision letter after peer review:**

Thank you for submitting your article "Species-specific sensitivity to TGFβ signaling and changes to the Mmp13 promoter underlie jaw development and evolution" for consideration by *eLife*. Your article has been reviewed by 3 peer reviewers, and the evaluation has been overseen by a Reviewing Editor and Kathryn Cheah as the Senior Editor. The reviewers have opted to remain anonymous.

The reviewers have discussed their reviews with one another, and the Reviewing Editor has drafted this to help you prepare a revised submission. The major required revisions are summarized below with further details provided in the full reviews.

Smith et al. examine jaw development across three different species of birds, chick, quail and duck, all of which have jaws of different shapes and sizes. They find that quail have higher TRAP activity and Mmp13 expression than duck in the jaw bones. By examining TGFβ signaling activity in vitro and in vivo, they show that quail and chick have higher and are more sensitive to TGFβ signaling activity than duck. They analyze the function of TGFβ signaling on its downstream target genes in these avian species. Interestingly, they find that two SNPs distinguish chick and quail from duck and that these two SNPs affect the differential species-specific response of the Mmp13 promoter. Taken together, this study provides interesting new data and insights into jaw development and evolution.

Essential revisions:

1. There should be more in vivo analysis to validate their findings. In addition, more evidence should be provided at cellular level to support their conclusions. For example, Mmp13 is also expressed in cartilage. Co-staining of Mmp13 with cartilage markers would help to strengthen the authors claim. It will be very helpful to validate the expression of the genes shown in Figure 2 in vivo because it is important to show the expression pattern of these genes and confirm where they are expressed in the jaw bones of these species. At least the genes that have significant changes at critical stages should be examined in vivo.

2. The result section in this manuscript should have more summary statements to help readers understand the context. In general, parts of the manuscript are difficult to read and would benefit from rewriting for clarity. The discussion rehashes the results to some extent.

3. Its current version mainly focuses on describing their outcomes while not providing integrated discussion. For example, the quail has the highest TGFβ signaling activity and duck has the lowest activity among the three species the authors have investigated. What is the impact of this on the jaw bone morphogenesis of these three species? This needs to be discussed in the manuscript. The TGFβ-Runx2-Mmp13 signaling axis seems very conservative across different species. This at least needs to be discussed in the manuscript. Previously published study has shown that mandible development is sensitive to the level of TGF signaling in mice. Together with this study, we can clearly see how important TGF signaling is in regulating jaw bone morphogenesis. Please add this into the discussion. It will be more impactful if the authors examined the TGFβ signaling activity in the upper jaw of these species. Developmental similarities between upper jaw and lower jaw need to be discussed in this manuscript.

4. In both quail and duck developing jaw, the authors show that region of bone resorption overlap with the domain of MMP13 expression. However, the extent of bone resorption is less severe in ducks and this correlates with a longer jaw length in duck compared to quail or chick. Based on this, the authors speculate that bone remodeling might play a crucial role in regulating jaw length. This result needs further justification. They did not analyze if jaw length is regulated during development in a similar manner in chick embryo. Such an analysis would strengthen the arguments in this manuscript.

5. Can the authors demonstrate that the identified sequence differences in the Mmp13 promoter affect SMAD or RUNX2 protein binding?

6. The manuscript is quite long and would profit from some rewriting and shortening.

*Reviewer #1 (Recommendations for the authors):*

1. Mmp13 is also expressed in cartilage. Co-staining of Mmp13 with cartilage markers would help to strengthen the authors' claim.

2. It would be very helpful to validate the expression of the genes shown in Figure 2 in vivo because it is important to show the expression pattern of these genes and confirm where they are expressed in the jaw bones of these species. At least the genes that have significant changes at critical stages should be examined in vivo.

3. The authors have shown Runx2 and Mmp13 are both downstream targets of TGFβ signaling in Figure 3 and Figure 4. Runx2 is important for bone formation during development and its expression is increased in response to TGFβ signaling. However, bone resorption is highlighted in this manuscript because Mmp13 expression is also increased. It is important to investigate whether Mmp13 has a direct effect on bone resorption in this context because TRAP activity is also elevated after TGFβ induction.

4. It seems the quail has the highest TGFβ signaling activity and duck has the lowest activity among the three species the authors have investigated. What is the impact of this on the jaw bone morphogenesis of these three species? This needs to be discussed in the manuscript. The TGFβ-Runx2-Mmp13 signaling axis seems to be highly conserved across different species. This at least needs to be discussed in the manuscript. Previously published studies have shown that mandible development is sensitive to the level of TGFβ signaling in mice.

5. It would be more impactful if the authors examine the TGFβ signaling activity in the upper jaw of these species. Developmental similarities between upper jaw and lower jaw need to be discussed in this manuscript.

6. The Results section in this manuscript should have more summary statements to help readers understand the context. The current version mainly focuses on describing the outcomes while not emphasizing the findings.

*Reviewer #2 (Recommendations for the authors):*

While I appreciate the evolutionary perspective of the work, the manuscript has a number of caveats. First it is quite long, and some sections are hard to get through (for instance the description of the data in Figures1 and 2 could be abridged). I also have the following additional comments:

1. In both quail and duck developing jaw, the authors show that region of bone resorption overlap with the domain of MMP13 expression. However, the extent of bone resorption is less severe in ducks and this correlates with a longer jaw length in duck compared to quail or chick. Based on this, the authors speculate that bone remodeling might play a crucial role in regulating jaw length. This result needs further justification. They did not analyze if jaw length is regulated during development in a similar manner in chick embryo. Such an analysis would strengthen the arguments in this manuscript.

2. Some of the data is very difficult to interpret/ follow. The authors point towards Figure 2 F-H Supplemental figure S3 F-G and Supplemental Figure S3B on page 22. In this case (A) Mmp2 and Pai1 expression analysis is missing; (B) MMP13 does not seem to be upregulated significantly (Supplementary Figure S3B); (C) Multiple data is not cited appropriately; for example, Supplemental figure S3 F-G, represents stimulation with recombinant TGFβ protein and has been inappropriately cited.

3. To analyze the sensitivity of TGFβ signaling pathway across species, the authors have compared chick and duck fibroblast cell lines, DF1 and CCL^-^141, respectively. For their analysis, since the jaw develops from neural crest mesenchymal cells, a more relevant cell line should have been used such as osteoblasts/ osteocytes.

4. Can the authors demonstrate that the identified sequence differences in the Mmp13 promoter affect SMAD or RUNX2 protein binding?

*Reviewer #3 (Recommendations for the authors):*

Here, I list just some changes, which could help readers to appreciate presented findings.

Introduction

Parts describing results of recent study can be shortened in introduction with summarizing just main results and not going through all findings.

While there is clear description how TGFβ activation leads to the induction of SMADs, there is missing information how exactly TGFβ activation controls the induction of Runx2 and how direct/indirect is this process. It would be useful for reader to summarize how exactly TGFβ targets Runx2 before going to results as this signaling is not so straightforward as in case of SMADs.

Results

Chapter "Bone resorption and MMP13 levels are species-specific and spatially regulated"

TRAP labeling surprisingly does not visualize osteoclasts very well. Are there differences in their number and distribution between species?

MMP13 labeling seems to display some unspecific labeling in several areas, it would be helpful to replace some of these pictures.

Result chapter "Sensitivity to TGFβ signaling is cell autonomous and species-specific" and "SNPs by a RUNX2 binding element affect the species-specific activity of Mmp13"

– These two chapters are very long, it would be useful to split them into several smaller subchapters focused on individual aims.

Discussion

Page 35: Authors mentioned „whereas elevated TGFβ signaling in Marfan syndrome causes excessive upper jaw growth". Would not be expected opposite effect based on presented results of the study? Can you explain such discrepancy?

Can you propose described process as a general mechanism also for other groups of vertebrates?

Are there expected some differences in mammals where Meckel cartilage is disrupted in development?

And how it is in case of endochondral bones where MMP13 is also a key factor of long bones ossification?

Discussion of such similarities or differences could help to reach broader audience.

---

## [Author Response]

Essential revisions:1. There should be more in vivo analysis to validate their findings. In addition, more evidence should be provided at cellular level to support their conclusions. For example, Mmp13 is also expressed in cartilage. Co-staining of Mmp13 with cartilage markers would help to strengthen the authors claim. It will be very helpful to validate the expression of the genes shown in Figure 2 in vivo because it is important to show the expression pattern of these genes and confirm where they are expressed in the jaw bones of these species. At least the genes that have significant changes at critical stages should be examined in vivo.

We have added more in vivo analyses. We have validated the expression of genes of interest (from Figure 2) by performing a new bulk RNAseq experiment (Figure 2—figure supplement 4). We measure expression in the developing jaw primordia of chick, quail, and duck to confirm our findings of species-specific differences in TGFβ pathway expression that we obtained via qPCR.

We have added new in situ hybridization (RNAscope) to examine the in vivo spatial expression of *Tgfβ1* and *Tgfβr1,* which are among the most differentially expressed ligands and receptors, in sections from chick, quail, and duck that are near adjacent to the ones used for histological analyses (Figure 1 and Figure 1—figure supplement 1).

We have added new IHC data to show that MMP13 is not expressed in any cartilage of the lower jaw skeleton at the stages we analyzed (Figure 1—figure supplement 1) and we have added the following text to explain this point:

“Although *Mmp13* is expressed by hypertrophic chondrocytes when cartilage is replaced by bone during endochondral ossification (Colnot and Helms, 2001), during development of the avian lower jaw, Meckel’s cartilage persists (*i.e.,* does not undergo hypertrophy) and there is no endochondral ossification except for that limited entirely to the most proximal region within the articular cartilage beginning after HH39 (Starck, 1989; Eames et al., 2004; Mitgutsch et al., 2011; Svandova et al., 2020). Therefore, as we have shown previously, *Mmp13* is not expressed in cartilage of the lower jaw skeleton (Ealba et al., 2015) nor do we detect MMP13 protein in the current study.”

2. The result section in this manuscript should have more summary statements to help readers understand the context. In general, parts of the manuscript are difficult to read and would benefit from rewriting for clarity. The discussion rehashes the results to some extent.

We have added summary statements to the subsections in the Results. We have re-written and edited the Introduction and Discussion sections for length, clarity, and content. We have removed the parts of the Discussion that were redundant with the Results section.

3. Its current version mainly focuses on describing their outcomes while not providing integrated discussion. For example, the quail has the highest TGFβ signaling activity and duck has the lowest activity among the three species the authors have investigated. What is the impact of this on the jaw bone morphogenesis of these three species? This needs to be discussed in the manuscript. The TGFβ-Runx2-Mmp13 signaling axis seems very conservative across different species. This at least needs to be discussed in the manuscript. Previously published study has shown that mandible development is sensitive to the level of TGF signaling in mice. Together with this study, we can clearly see how important TGF signaling is in regulating jaw bone morphogenesis. Please add this into the discussion. It will be more impactful if the authors examined the TGFβ signaling activity in the upper jaw of these species. Developmental similarities between upper jaw and lower jaw need to be discussed in this manuscript.

We have rewritten and reorganized the manuscript so that the Discussion and Conclusion are more integrated thematically and so that we can explain the implications of each of our findings. We have added discussion throughout about the impact of this developmental mechanism on jaw bone morphogenesis (specifically bone deposition and resorption during osteogenesis) in these different species. We have added more discussion on the conservation of the TGFβ-Runx2-Mmp13 signaling axis and how the regulatory changes observed in our study provide a novel insight in to how the pathway can evolve over time and contribute to different morphological outcomes*.* We have added to the Discussion more details and citations to the work in mice and human disease that shown that mandible development is sensitive to the level of TGFβ signaling. While we have not added data on the upper jaw because this was not experimentally feasible given the constraints described above, we have added a discussion of the developmental similarities between the upper and lower jaw with regard to TGFβ signaling.

4. In both quail and duck developing jaw, the authors show that region of bone resorption overlap with the domain of MMP13 expression. However, the extent of bone resorption is less severe in ducks and this correlates with a longer jaw length in duck compared to quail or chick. Based on this, the authors speculate that bone remodeling might play a crucial role in regulating jaw length. This result needs further justification. They did not analyze if jaw length is regulated during development in a similar manner in chick embryo. Such an analysis would strengthen the arguments in this manuscript.

We have added more details on our prior work showing that if we block bone resorption or MMP13 then we can lengthen the jaw in quail (Ealba et al. 2015) to the Discussion. We have also added a new *Mmp13* overexpression experiment in the developing lower jaw of duck. We employed our stably-integrating and dox-inducible overexpression construct (Chu et al., 2020), which we electroporated into presumptive neural crest mesenchyme of duck at HH8.5. We induced embryos with dox at HH34 and collected specimens at HH40. We validated the enzymatic activity of our construct using an EnzChek assay kit, we confirmed overexpression in neural crest mesenchyme via RFP, and we assay for changes in jaw length through µCT and morphometric analyses (Figure 5 and Figure 5—figure supplement 1). We find that *Mmp13* overexpression significantly shortens the lower jaw in duck, which along with our previously published inhibition experiments in quail, demonstrates that avian jaw length can be modulated by *Mmp13* expression and bone resorption. We have also added new histological data on bone resorption (TRAP), protein expression (MMP13 IHC), and gene expression on the TGFβ pathway (qPCR, RNAseq, and RNAscope) in chick embryos to the Results section (Figure 1, Figure 1—figure supplement 1 and Figure 2—figure supplement 4).

5. Can the authors demonstrate that the identified sequence differences in the Mmp13 promoter affect SMAD or RUNX2 protein binding?

We have added new electrophoretic mobility shift assays (EMSA) to detect protein–nucleic acid interactions (Figure 7 and Figure 7—figure supplement 1). We performed EMSA with *Mmp13* promoter oligos containing the SMAD binding element, the RUNX2 binding element, and without the binding elements in combination with SMAD4 recombinant protein or *Runx2* overexpression. Our results demonstrate these promoter elements are critical for SMAD and RUNX2 binding in the *Mmp13* promoter.

6. The manuscript is quite long and would profit from some rewriting and shortening.

As described above, we have re-written and edited the Introduction and Discussion sections for length, clarity, and content. We have removed parts of the Discussion that were redundant with the Results section.

Reviewer #1 (Recommendations for the authors):1. Mmp13 is also expressed in cartilage. Co-staining of Mmp13 with cartilage markers would help to strengthen the authors' claim.

While Reviewer 1 is correct that *Mmp13* is also expressed in cartilage, this mostly pertains to cartilages of the axial and appendicular skeleton, as well as the cranial base and sensory capsules that undergo endochondral ossification via hypertrophy of a cartilaginous template. The only cartilage in the avian lower jaw is Meckel’s cartilage, and Meckel’s is a persistent cartilage that does not get replaced by bone nor does Meckel’s express *Mmp13*. In response to the Reviewer’s point, we have added new IHC data to show that MMP13 is not expressed in Meckel’s cartilage of the lower jaw skeleton at the stages we analyzed (Figure 1—figure supplement 1) and we have added the following text to explain this point:

“Although *Mmp13* is expressed by hypertrophic chondrocytes when cartilage is replaced by bone during endochondral ossification (Colnot and Helms, 2001), during development of the avian lower jaw, Meckel’s cartilage persists (*i.e.,* does not undergo hypertrophy) and there is no endochondral ossification except for that limited entirely to the most proximal region within the articular cartilage beginning after HH39 (Starck, 1989; Eames et al., 2004; Mitgutsch et al., 2011; Svandova et al., 2020). Therefore, as we have shown previously, *Mmp13* is not expressed in cartilage of the lower jaw skeleton (Ealba et al., 2015) nor do we detect MMP13 protein in the current study.”

2. It would be very helpful to validate the expression of the genes shown in Figure 2 in vivo because it is important to show the expression pattern of these genes and confirm where they are expressed in the jaw bones of these species. At least the genes that have significant changes at critical stages should be examined in vivo.

We agree and have added an in situ hybridization (*i.e.,* RNAscope) experiment showing the presence of a ligand (*i.e., Tgfβ1*) and a receptor (*i.e., Tgfβr1*) in bone sections near adjacent to our histology, since these showed significant amounts of differential expression between quail and duck in our analysis. To provide additional independent confirmation, we have also added data from an RNAseq experiment that corroborates the qPCR data on differential expression of TGFβ pathway members.

3. The authors have shown Runx2 and Mmp13 are both downstream targets of TGFβ signaling in Figure 3 and Figure 4. Runx2 is important for bone formation during development and its expression is increased in response to TGFβ signaling. However, bone resorption is highlighted in this manuscript because Mmp13 expression is also increased. It is important to investigate whether Mmp13 has a direct effect on bone resorption in this context because TRAP activity is also elevated after TGFβ induction.

We have added more details on our prior work showing that if we block bone resorption or MMP13 then we can lengthen the jaw in quail (Ealba et al. 2015) to the Introduction and Discussion. We have also added a new *Mmp13* overexpression experiment in the developing lower jaw of duck. We employed our stably-integrating and dox-inducible overexpression construct (Chu et al., 2020), which we electroporated into presumptive neural crest mesenchyme of duck at HH8.5. We induced embryos with dox at HH34 and collected specimens at HH40. We validated the enzymatic activity of our construct using an EnzChek assay kit, we confirmed overexpression in neural crest mesenchyme via RFP, and we assay for changes in jaw length through µCT and morphometric analyses (Figure 5 and Figure 5—figure supplement 1). We find that *Mmp13* overexpression significantly shortens the lower jaw in duck, which along with our previously published inhibition experiments in quail, demonstrates that avian jaw length can be modulated by *Mmp13* expression and bone resorption.

4. It seems the quail has the highest TGFβ signaling activity and duck has the lowest activity among the three species the authors have investigated. What is the impact of this on the jaw bone morphogenesis of these three species? This needs to be discussed in the manuscript. The TGFβ-Runx2-Mmp13 signaling axis seems to be highly conserved across different species. This at least needs to be discussed in the manuscript. Previously published studies have shown that mandible development is sensitive to the level of TGFβ signaling in mice.

We have added more discussion on the conservation of the TGFβ-Runx2-Mmp13 signaling axis and how the regulatory changes observed in our study provide a novel insight in to how the pathway can evolve over time and contribute to different morphological outcomes*.* We have added to the Discussion more details and citations to the work in mice and human disease that shown that mandible development is sensitive to the level of TGFβ signaling.

5. It would be more impactful if the authors examine the TGFβ signaling activity in the upper jaw of these species. Developmental similarities between upper jaw and lower jaw need to be discussed in this manuscript.

We have added more discussion on the conservation of the TGFβ-Runx2-Mmp13 signaling axis and how the regulatory changes observed in our study provide a novel insight in to how the pathway can evolve over time and contribute to different morphological outcomes*.* We have added to the Discussion more details and citations to the work in mice and human disease that shown that mandible development is sensitive to the level of TGFβ signaling.

6. The Results section in this manuscript should have more summary statements to help readers understand the context. The current version mainly focuses on describing the outcomes while not emphasizing the findings.

We have added summary statements to the subsections in the Results. We have rewritten and reorganized the manuscript so that the Discussion and Conclusion are more integrated thematically and so that we can explain the implications of each of our findings.

Reviewer #2 (Recommendations for the authors):While I appreciate the evolutionary perspective of the work, the manuscript has a number of caveats. First it is quite long, and some sections are hard to get through (for instance the description of the data in Figures1 and 2 could be abridged). I also have the following additional comments:

We have re-written and edited each of the manuscript sections for length, clarity, and content. We have removed parts of the Discussion that were redundant with the Results section.

1. In both quail and duck developing jaw, the authors show that region of bone resorption overlap with the domain of MMP13 expression. However, the extent of bone resorption is less severe in ducks and this correlates with a longer jaw length in duck compared to quail or chick. Based on this, the authors speculate that bone remodeling might play a crucial role in regulating jaw length. This result needs further justification. They did not analyze if jaw length is regulated during development in a similar manner in chick embryo. Such an analysis would strengthen the arguments in this manuscript.

As described above, we have added more details on our prior work showing that if we block bone resorption or MMP13 then we can lengthen the jaw in quail (Ealba et al. 2015) to the Introduction and Discussion. We have also added a new *Mmp13* overexpression experiment in the developing lower jaw of duck, which significantly shortens the lower jaw. This along with our previously published inhibition experiments in quail, demonstrates that avian jaw length can be modulated by *Mmp13* expression and bone resorption. We have added new TRAP and MMP13 data for chick to our study showing that expression levels are quail-like.

2. Some of the data is very difficult to interpret/ follow. The authors point towards Figure 2 F-H Supplemental figure S3 F-G and Supplemental Figure S3B on page 22. In this case (A) Mmp2 and Pai1 expression analysis is missing; (B) MMP13 does not seem to be upregulated significantly (Supplementary Figure S3B); (C) Multiple data is not cited appropriately; for example, Supplemental figure S3 F-G, represents stimulation with recombinant TGFβ protein and has been inappropriately cited.

We sincerely apologize for this and have tried to fix all these errors/oversights.

3. To analyze the sensitivity of TGFβ signaling pathway across species, the authors have compared chick and duck fibroblast cell lines, DF1 and CCL^-^141, respectively. For their analysis, since the jaw develops from neural crest mesenchymal cells, a more relevant cell line should have been used such as osteoblasts/ osteocytes.

Unfortunately, there are no chick, quail, or duck osteoblast or osteocyte cell lines available.

Since DF-1 and CCL^-^141 are fibroblast cell lines derived from developing embryos, we feel they are useful in our experiments to address certain types of questions that we can then also test in ovo. Most importantly, these cell culture experiments allow us to assess the effects of intrinsic/hard-wired genetic changes that arose during evolution (presumably almost 100 million years ago) and are present in the genomes of chick and duck cells regardless of cell type or source; and they enable us to evaluate the extent to which these effects are cell-autonomous or context dependent.

4. Can the authors demonstrate that the identified sequence differences in the Mmp13 promoter affect SMAD or RUNX2 protein binding?

We have added EMSA that show interactions between SMAD4 and the SMAD binding elements on the *Mmp13* promoter (Figure 7 and Figure 7—figure supplement 1). We have also added EMSA that show interactions between RUNX2 and the RUNX2 binding elements on the *Mmp13* promoter (Figure 7—figure supplement 1).

Reviewer #3 (Recommendations for the authors):Here, I list just some changes, which could help readers to appreciate presented findings.IntroductionParts describing results of recent study can be shortened in introduction with summarizing just main results and not going through all findings.

We have shortened the description of our results in the Introduction.

While there is clear description how TGFβ activation leads to the induction of SMADs, there is missing information how exactly TGFβ activation controls the induction of Runx2 and how direct/indirect is this process. It would be useful for reader to summarize how exactly TGFβ targets Runx2 before going to results as this signaling is not so straightforward as in case of SMADs.

We have included citations and some details to the Introduction on how TGFβ activation leads to activation of SMADs, which in turn regulate *Runx2*. We also discuss how *Mmp13* can be regulated by TGFβ signaling not only through SMADs but through upregulation of *Runx2* as well.

ResultsChapter "Bone resorption and MMP13 levels are species-specific and spatially regulated"TRAP labeling surprisingly does not visualize osteoclasts very well. Are there differences in their number and distribution between species?

Yes, we analyzed osteoclasts (and osteocytes) in our previously published work (Ealba et al. 2015) where we quantified TRAP staining in quail, duck, and quail-duck chimeras (“quck”). There are species-specific differences, and these are mediated by neural crest mesenchyme.

MMP13 labeling seems to display some unspecific labeling in several areas, it would be helpful to replace some of these pictures.

Unfortunately, commercial MMP13 antibodies are not available for any of the avian species we studied. Therefore, we had a custom antibody made for MMP13. Due to this limitation, there may be unspecific labeling in limited regions of the tissue, but the signal is clearly the strongest in the bone, which correlates with our previously published data (Ealba et al. 2015) showing *Mmp13* gene expression highest in these bones as well. We do not observe MMP13 protein in the cartilage of the lower jaw (Meckel’s) but we do observe additional labeling in the epidermis, which is known to express *Mmp13*.

Result chapter "Sensitivity to TGFβ signaling is cell autonomous and species-specific" and"SNPs by a RUNX2 binding element affect the species-specific activity of Mmp13"– These two chapters are very long, it would be useful to split them into several smaller subchapters focused on individual aims.

We have tried wherever possible to shorten the text.

DiscussionPage 35: Authors mentioned „whereas elevated TGFβ signaling in Marfan syndrome causes excessive upper jaw growth". Would not be expected opposite effect based on presented results of the study? Can you explain such discrepancy?

This is an excellent point. We have included in the paper a discussion as to how the timing and levels of expression of TGFβ signaling is important when considering any interpretation of phenotypic outcomes. For example, TGFβ signaling plays very different roles during osteoblast differentiation, at first promoting osteoblast differentiation, proliferation, and upregulation of Runx2, whereas later TGFβ signaling inhibits osteoblast differentiation by suppressing Runx2 expression. TGFβ signaling is highly complex and is highly integrated into other pathways as well, which can make broad interpretations of its effect challenging. Marfan syndrome may take place through an entirely different signaling axis then what we observe in our paper.

Can you propose described process as a general mechanism also for other groups of vertebrates?

We have now included a discussion of how the regulatory changes we observe may have broad implications for morphological evolution.

Are there expected some differences in mammals where Meckel cartilage is disrupted in development?

Great question. Yes, we would expect there to be differences in jaw size where Meckel’s cartilage is disrupted. However, our results suggest that the TGFB-RUNX2-MMP13 axis may not be the primary driver of size control in Meckel’s during development, especially at the later stages that we examined. We would expect birds and mammals to differ since the maturation process for Meckel’s is distinct. We have described the condition for birds in the manuscript and the condition for mammals is thoroughly described elsewhere.

And how it is in case of endochondral bones where MMP13 is also a key factor of long bones ossification?

As described above, there are no bones that form through endochondral ossification in the avian lower jaw (except for the most proximal region that forms the articular bone starting after HH39). While this is a fascinating topic, we feel it is beyond the scope of the present study and too much to add when we are trying to shorter the manuscript already.